# EMP: Effective Multidimensional Persistence for Graph Representation Learning

## Abstract

Topological data analysis (TDA) has become increasingly popular in a broad range of machine learning tasks, ranging from anomaly detection and manifold learning to graph classification. Persistent homology being the key approach in TDA provides a unique topological fingerprint of the data by assessing the evolution of various hidden patterns in the data as we vary a scale parameter. Current PH tools are limited to analyze the data by filtering with single parameter while in many applications, several relevant parameters are equally important to get a much finer information on the data. In this paper, we overcome this problem by introducing Effective Multidimensional Persistence (EMP) framework which enables to investigate the data by varying multiple scale parameters simultaneously. EMP framework provides a highly expressive summary of the data by integrating the multiple descriptor functions to the process successfully. EMP naturally adapts to many known single PH summaries and converts them into multidimensional summaries, for example, EMP Landscapes, EMP Silhouettes, EMP Images, and EMP Surfaces. These summaries deliver a multidimensional fingerprint of the data as matrices and arrays which are suitable for various machine learning models. We apply EMP framework in graph classification tasks and observe that EMP boosts the performances of various single PH descriptors, and outperforms the most state-of-the-art methods on benchmark datasets. We further derive theoretical guarantees of the proposed EMP summary and prove the stability properties.

## 1 Introduction

In the past decade, topological data analysis (TDA) proved to be a powerful machinery to discover many hidden patterns in various forms of data which are otherwise inaccessible with more traditional methods. In particular, for graph machine learning tasks, while many traditional methods fail, TDA and, specifically, tools of persistent homology (PH), have demonstrated a high potential to detect local and global patterns and to produce a unique topological fingerprint to be used in various machine learning tasks. This makes PH particularly attractive for capturing various characteristics of the complex data which may play the key role behind the learning task performance.

In turn, multiparameter persistence, or multipersistence (MP) is a novel idea to further advance the PH machinery by analysing the data in much finer way simultaneously along multiple dimensions. However, because of the technical problems related to commutative algebra because of its multidimensional structure, it has not been defined for general settings yet (See Section 2.1). In this paper, we develop an alternative approach to utilize multipersistence idea very efficiently for various types of data, with main focus on graph representation. In particular, we bypass technical issues with the MP by obtaining a very practical summaries by utilizing slicing idea in a structured way. In turn, we obtain suitable multidimensional topological fingerprints of the data as matrices and arrays where ML models can easily detect the hidden patterns developed in the complex data.

**Our contributions** can be summarized as follows:

- We develop a new computationally efficient and highly expressive EMP framework which provides multidimensional topological fingerprints of the data. EMP expands many popular summaries of single persistence to multidimensions by adapting an effective slicing

direction. As such, our EMP framework provides a practical way to utilize the promising multipersistence approach in real-life applications.

- We illustrate the utility of EMP summaries in various settings and compare our results to state-of-the-art methods. Our numerical experiments demonstrate that EMP summaries outperforms SOTA in several benchmark datasets for graph classification tasks.

- We derive theoretical stability guarantees of the new topological summaries.

## 2 RELATED WORK

### 2.1 MULTIPARAMETER PERSISTENCE

Persistent homology (PH), being a key tool in topological data analysis (TDA), delivers invaluable and complementary information on the intrinsic properties of data that are inaccessible with conventional methods (Chazal & Michel, 2021; Hensel et al., 2021). In the past decade, PH has become quite popular in various ML tasks, ranging from manifold learning to medical image analysis, material science to finance (TDA applications library (Giunti, 2022)). One of the key benefits of PH is that it allows us to extract the evolution of subtler patterns in the data shape dynamics at multiple resolution scales which are not accessible with more conventional, non-topological methods (Wasserman, 2018).

Multipersistence (MP) is a highly promising approach to significantly improve the success of single parameter persistence (SP) in applied topological data analysis, but there are some issues to convert this novel idea into an effective feature extraction method as desired (See Appendix D.3). Except for some special cases, MP theory suffers from the problem of the nonexistence of barcode decomposition because of the partially ordered structure of the index set $\{(\alpha_i, \beta_j)\}$ (Botnan & Lesnick, 2022). Lesnick & Wright (2015) suggested to bypass this issue via *slicing technique* by studying one-dimensional fibers of the multiparameter domain where one restricts the multidimensional persistence module to a single direction (slice) and to use single persistence on this one dimensional slice. Later, by using this novel idea, Carrière & Blumberg (2020) combined several slicing directions (vineyards) and obtained a vectorization by summarizing the persistence diagrams (PDs) in these directions. There are several promising recent studies in this direction (Botnan et al., 2021; Vipond, 2020), but these approaches are not computationally feasible, and cannot provide the expected effectiveness of MP approach in real life applications. Here we develop a highly efficient way to use MP approach for various forms of data, and provide a multidimensional topological vectorization with EMP summaries.

### 2.2 GRAPH REPRESENTATION LEARNING

After the success of convolutional neural networks (CNN) on image-based tasks, graph neural networks (GNNs) have emerged as a powerful tool for graph-level classification and representation learning. A wide variety of models are developed based on numerous theories (see Appendix A for further details). However, to our best knowledge, most of existing approaches do not account for the important topological information on the shapes of the node neighborhods. While GNNs produce great performances in many graph learning tasks, they tend to suffer from over-smoothing problems and are vulnerable to graph perturbations. To address these challenges, TDA provides a computationally efficient alternative to GNNs, and can be used as an effective feature extractor to be combined with the deep learning methods Hofer et al. (2020); Bodnar et al. (2022); Horn et al. (2021). Recently, this direction has been explored in many works, and single persistent homology give highly competitive results with kernel-based methods Togninalli et al. (2019); Rieck et al. (2019); Le & Yamada (2018); Zhao & Wang (2019); Kyriakis et al. (2021) and neural networks Hofer et al. (2019); Carrière et al. (2020) in graph classification tasks. In this work, we apply MP approach for the first time in this setting, and our EMP model outperforms most deep learning models in benchmark datasets.

## 3 BACKGROUND

We start from providing the basic background for our framework. Since we mainly focus on graph representation learning in this paper, we explain our construction on graph setting. Note that

our techniques can easily be adapted to other types of data, e.g., point clouds and images (See Appendix D.1).

## 3.1 PERSISTENT HOMOLOGY

Persistent Homology (PH) is a mathematical machinery to capture the hidden shape patterns in the data by using algebraic topology tools. PH extracts this information by keeping track of evolution of the topological features (components, loops, cavities) created in the data while looking at it in different resolutions Dey & Wang (2022). Appendix B provides additional background on classic PH for the avid reader.

For a given graph $\mathcal{G}$, consider a nested sequence of subgraphs $\mathcal{G}_1 \subseteq \ldots \subseteq \mathcal{G}_N = \mathcal{G}$. For each $\mathcal{G}_i$, define an abstract simplicial complex $\widehat{\mathcal{G}}_i$, $1 \leq i \leq N$, yielding a filtration of complexes $\widehat{\mathcal{G}}_1 \subseteq \ldots \subseteq \widehat{\mathcal{G}}_N$. Here, clique complexes are among the most common ones, i.e., a clique complex $\widehat{\mathcal{G}}$ is obtained by assigning (filling with) a $k$-simplex to each complete $(k + 1)$-complete subgraph in $\mathcal{G}$, e.g., a 3-clique, a complete 3-subgraph, in $\mathcal{G}$ will be filled with a 2-simplex (triangle). Then, in this sequence of simplicial complexes, one can systematically keep track of the evolution of topological patterns mentioned above. A $k$-dimensional topological feature (or $k$-hole) may represent connected components (0-hole), loops (1-hole) and cavities (2-hole). For each $k$-hole $\sigma$, PH records its first appeareance in the filtration sequence, say $\widehat{\mathcal{G}}_{b_\sigma}$, and first disappearence in later complexes, $\widehat{\mathcal{G}}_{d_\sigma}$ with a unique pair $(b_\sigma, d_\sigma)$, where $1 \leq b_\sigma < d_\sigma \leq N$. We call $b_\sigma$ *the birth time* of $\sigma$ and $d_\sigma$ *the death time* of $\sigma$. We call $d_\sigma - b_\sigma$ *the life span* of $\sigma$. PH records all these birth and death times of the topological features in *persistence diagrams* (PD). Let $0 \leq k \leq D$ where $D$ is the highest dimension in the simplicial complex $\widehat{\mathcal{G}}_N$. Then the $k$th persistence diagram $\mathrm{PD}_k(\mathcal{G}) = \{(b_\sigma, d_\sigma) \mid \sigma \in H_k(\widehat{\mathcal{G}}_i) \text{ for } b_\sigma \leq i < d_\sigma\}$. Here, $H_k(\widehat{\mathcal{G}}_i)$ represents the $k^{th}$ homology group of $\widehat{\mathcal{G}}_i$ which keeps the information of the $k$-holes in the simplicial complex $\widehat{\mathcal{G}}_i$. For sake of notation, we skip the dimension (subscript $k$). With the intuition that the topological features with long life span (persistent features) describe the hidden shape patterns in the data, these PDs provide a unique topological fingerprint of $\mathcal{G}$.

As one can easily notice, the most important step in the PH machinery is the construction of the nested sequence of subgraphs $\mathcal{G}_1 \subseteq \ldots \subseteq \mathcal{G}_N = \mathcal{G}$. For a given unweighted graph $\mathcal{G} = (\mathcal{V}, \mathcal{E})$ with $\mathcal{V} = \{v_1, \ldots, v_{\mathcal{N}}\}$ the set of nodes and $\mathcal{E} \subset \{\{v_i, v_j\} \in \mathcal{V} \times \mathcal{V}, i \neq j\}$ the set of edges, the most common technique is to use a filtration function $f : \mathcal{V} \to \mathbb{R}$ with a choice of thresholds $\mathcal{I} = \{\alpha_i\}$ where $\alpha_1 = \min_{v \in \mathcal{V}} f(v) < \alpha_2 < \ldots < \alpha_N = \max_{v \in \mathcal{V}} f(v)$. For $\alpha_i \in \mathcal{I}$, let $\mathcal{V}_i = \{v_r \in \mathcal{V} \mid f(v_r) \leq \alpha_i\}$. Let $\mathcal{G}_i$ be the induced subgraph of $\mathcal{G}$ by $\mathcal{V}_i$, i.e., $\mathcal{G}_i = (\mathcal{V}_i, \mathcal{E}_i)$ where $\mathcal{E}_i = \{e_{rs} \in \mathcal{E} \mid v_r, v_s \in \mathcal{V}_i\}$. This process yields a nested sequence of subgraphs $\mathcal{G}_1 \subset \mathcal{G}_2 \subset \ldots \subset \mathcal{G}_N = \mathcal{G}$, called *the sublevel filtration* induced by the filtration function $f$. We denote PDs obtained via sublevel filtration for a filtration function $f$ as $PD(\mathcal{G}, f)$. The choice of $f$ is crucial here, and in most cases, $f$ is either an important function from the domain of the data, e.g., atomic number in protein or a function defined from intrinsic properties of the graph, e.g., degree and betweenness. Similarly, for a weighted graph, one can use sublevel filtration on the weights of the edges and obtain a suitable filtration reflecting the domain information stored in the edge weights. (For further details on different filtrations see Aktas et al. (2019); Hofer et al. (2020).)

## 3.2 SINGLE PERSISTENCE VECTORIZATIONS

While PH extracts hidden shape patterns from data as persistence diagrams (PD), PDs being collection of points in $\mathbb{R}^2$ by itself are not very practical for statistical and machine learning purposes. Instead, the common techniques are by faithfully representing PDs as kernels Kriege et al. (2020) or vectorizations Hensel et al. (2021). This provides a practical way to use the outputs of PH in real life applications. *Single Persistence Vectorizations* transform obtained PH information (PDs) into a function or a feature vector form which are much more suitable for machine learning tools than PDs. Common single persistence (SP) vectorization methods are Persistence Images Adams et al. (2017), Persistence Landscapes Bubenik (2015), Silhouettes Chazal et al. (2014), and various Persistence Curves Chung & Lawson (2019). These vectorizations defines a single variable or multivariable functions out of PDs, which can be used as fixed size $1D$ or $2D$ vectors in applications, i.e $1 \times n$ vectors or $m \times n$ vectors. For example, a Betti curve for a PD with $n$ thresholds can also be expressed

as $1 \times n$ size vectors. Similarly, Persistence Images is an example of $2D$ vectors with the chosen resolution (grid) size. See the examples given in Section 4.2 for further details.

## 3.3 MULTIDIMENSIONAL PERSISTENCE

MultiPersistence (MP) is a novel idea to significantly boost the performance of the single parameter persistence technique described above. The reason for the term "single" is that we are filtering the data in only one direction $\mathcal{G}_1 \subset \cdots \subset \mathcal{G}_N = \mathcal{G}$. The construction of the filtration is the key to get fine analysis of the data to find the hidden patterns. If one uses one function $f : \mathcal{V} \to \mathbb{R}$ which has very valuable domain information (e.g., amount for blockchain networks, atomic number for protein networks), then this induces a single parameter filtration as above. However, various data have more than one very natural domain functions to analyze the data, and using them simultaneously would give a much better understanding of the hidden patterns. With this intuition, multiparameter persistence (MP) theory is suggested as natural generalization of single persistence (SP).

In simple terms, if one uses only one filtration function, sublevel sets induces a single parameter filtration $\widehat{\mathcal{G}}_1 \subset \cdots \subset \widehat{\mathcal{G}}_N = \widehat{\mathcal{G}}$. Instead, if one uses two or more functions, then it would give a way to study the data in much finer resolution. For example, if we have two functions $f : \mathcal{V} \to \mathbb{R}$ and $g : \mathcal{V} \to \mathbb{R}$ with very valuable complementary information of the network, MP idea is presumed to produce a unique topological fingerprint combining the information from both functions. These pair of functions $f, g$ induces a multivariate filtration function $F : \mathcal{V} \to \mathbb{R}^2$ with $F(v) = (f(v), g(v))$. Again, one can define a set of nondecreasing thresholds $\{\alpha_i\}_1^m$ and $\{\beta_j\}_1^n$ for $f$ and $g$ respectively. Then, $\mathcal{V}_{ij} = \{v_r \in \mathcal{V} \mid f(v_r) \leq \alpha_i, g(v_r) \leq \beta_j\}$, i.e., $\mathcal{V}_{ij} = F^{-1}((-\infty, \alpha_i] \times (-\infty, \beta_j])$. Then, let $\mathcal{G}_{ij}$ be the induced subgraph of $\mathcal{G}$ by $\mathcal{V}_{ij}$, i.e., the smallest subgraph of $\mathcal{G}$ generated by $\mathcal{V}_{ij}$. Then, instead of a single filtration of complexes, we get a bifiltration of complexes $\{\widehat{\mathcal{G}}_{ij} \mid 1 \leq i \leq m, 1 \leq j \leq n\}$. One can imagine $\{\widehat{\mathcal{G}}_{ij}\}$ as a rectangular grid of size $m \times n$ such that for each $1 \leq i_0 \leq m$, $\{\widehat{\mathcal{G}}_{i_0 j}\}_{j=1}^n$ gives a nested (horizontal) sequence of simplicial complexes. Similarly, for each $1 \leq j_0 \leq n$, $\{\widehat{\mathcal{G}}_{i j_0}\}_{i=1}^m$ gives a nested (vertical) sequence of simplicial complexes (See Figure 2 in the Appendix).

By computing the homology groups of these complexes, $\{H_k(\widehat{\mathcal{G}}_{ij})\}$, we obtain the induced bigraded persistence module (a rectangular grid of size $m \times n$). Again, the idea is to keep track of the $k$-dimensional topological features via the homology groups $\{H_k(\widehat{\mathcal{G}}_{ij})\}$ in this grid. As explained in Appendix D.3, because of the technical issues related to commutative algebra, there are still issues to define a mathematical output like "Multipersistence Diagram", and hence, to obtain an effective vectorization of the MP module (Section 3). These technical issues prevent this promising approach to reach its full potential in real life applications.

In this paper, we overcome this problem by producing very practical vectorizations by utilizing slicing idea in the multipersistence grid in a structured way. The key point in our approach is that we produce *multidimensional vectors* (matrices and arrays) as unique topological multidimensional fingerprints of the data. In other words, by using MP approach, we extract multidimensional features of the data as same size matrices and arrays which are quite suitable for various machine learning models, and we can facilitate the appropriate ML method to analyze these unique fingerprints and detect the patterns developing in these multi-resolution view of the data.

## 4 EFFECTIVE MULTIDIMENSIONAL PERSISTENCE SUMMARIES

We now introduce our Effective MultiPersistence (EMP) framework which describes a way to expand most single persistence vectorizations (Section 3.2) as multidimensional vectorizations by utilizing MP approach. In particular, by using the existing single parameter persistence vectorizations, we produce multidimensional vectorization by effectively using the one of the filtering direction as "slicing direction" in the multipersistence module.

### 4.1 EMP FRAMEWORK

In simple terms, for $d = 2$, for a given two filtration functions $f, g$, the main idea is to use first filtration function $f$ to get a single parameter filtering of the data, i.e., $\mathcal{G}_1 \subseteq \ldots \subseteq \mathcal{G}_m = \mathcal{G}$. Then,

use the second function in each subgraph to obtain persistence diagram $PD(\mathcal{G}_i, g)$ for $1 \leq i \leq m$. In particular, we obtain $m$ persistence diagrams $\{PD(\mathcal{G}_i, g)\}_{i=1}^m$. Then, by applying the chosen SP vectorization $\varphi$ to each PD, we obtain $m$ different same length vector $\vec{\varphi}(PD(\mathcal{G}_i, g))$, say $1 \times k$ vector. Then, by combining all $m$ $1D$-vectors, we obtain EMP vectorization $\mathbf{M}_\varphi$ such that $\mathbf{M}_\varphi^i = \vec{\varphi}(PD(\mathcal{G}_i, g))$ where $\mathbf{M}_\varphi^i$ represents the $i^{th}$ row of $\mathbf{M}_\varphi$ which is a $2D$-vector (matrix) of size $m \times k$ (See Figure 3 in Appendix). For $d > 2$, the construction is similar and given below.

Here, we give the details for sublevel filtration for two filtration functions. In Appendix D.2, we explain how to modify the construction for weight filtrations or power filtrations. Let $\mathcal{G} = (\mathcal{V}, \mathcal{E})$ be a graph, and let $f, g : \mathcal{V} \to \mathbb{R}$ be two filtration functions with threshold sets $\{\alpha_i\}_{i=1}^m$ and $\{\beta_j\}_{j=1}^n$ respectively. Let $\mathcal{V}_i = \{v_r \in \mathcal{V} \mid f(v_r) \leq \alpha_i\}$. Let $\mathcal{G}_i$ be the induced subgraph of $\mathcal{G}$ by $\mathcal{V}_i$. This gives a filtering of the graph (nested subgraphs) as $\mathcal{G}_1 \subseteq \ldots \subseteq \mathcal{G}_m = \mathcal{G}$. Recall that $g : \mathcal{V} \to \mathbb{R}$ is another filtration function for $\mathcal{G}$. Now, we fix $1 \leq i_0 \leq m$, and consider $\mathcal{G}_{i_0}$. By restricting $g$ on $\mathcal{G}_{i_0}$, we get persistence diagram $PD(\mathcal{G}_{i_0}, g)$ as follows. Let $\mathcal{V}_{i_0 j} = \{v_r \in \mathcal{V}_{i_0} \mid f(v_r) \leq \beta_j\}$, and let $\mathcal{G}_{i_0 j}$ be the induced subgraph of $\mathcal{G}_{i_0}$ by $\mathcal{V}_{i_0 j}$. This defines a finer filtering of the graph $\mathcal{G}_{i_0}$ as $\mathcal{G}_{i_0 1} \subseteq \mathcal{G}_{i_0 2} \ldots \subseteq \mathcal{G}_{i_0 n} = \mathcal{G}_{i_0}$. Corresponding clique complexes defines a filtration $\widehat{\mathcal{G}}_{i_0 1} \subseteq \widehat{\mathcal{G}}_{i_0 2} \ldots \subseteq \widehat{\mathcal{G}}_{i_0 n} = \widehat{\mathcal{G}}_{i_0}$. This filtration gives the persistence diagram $PD(\mathcal{G}_{i_0}, g)$. Hence, for each $1 \leq i \leq m$, we obtain a persistence diagram $PD(\mathcal{G}_i, g)$.

The next step is to use the vectorization on these $m$ persistence diagrams. Let $\varphi$ be a single persistence vectorization, e.g., Persistence Landscape, Silhouette, Entropy, Betti, Persistence Image. By applying the chosen SP vectorization $\varphi$ to each PD, we obtain a function $\varphi_i = \varphi(PD(\mathcal{G}_i, g))$ where in most cases it is a single variable function on the threshold domain $[\beta_1, \beta_n]$, i.e., $\varphi_i : [\beta_1, \beta_n] \to \mathbb{R}$. For the multivariable case (e.g., Persistence Image), we give explicit description in the examples section below. Most such vectorizations being induced from a discrete set of points $PD(\mathcal{G})$, they naturally can be expressed as a $1D$ vector of length $k$. In the examples below, we explain this conversion in detail. Then, let $\vec{\varphi}_i$ be the corresponding $1 \times k$ vector for the function $\varphi_i$. Now, we are ready to define our EMP summary $\mathbf{M}_\varphi$ which is a $2D$-vector (a matrix)

$$\mathbf{M}_\varphi^i = \vec{\varphi}_i \quad \text{for} \quad 1 \leq i \leq m,$$

where $\mathbf{M}_\varphi^i$ is the $i^{th}$-row of $\mathbf{M}_\varphi$. Hence, $\mathbf{M}_\varphi$ is a $2D$-vector of size $m \times k$. Each row $\mathbf{M}_\varphi^i$ is the vectorization of the persistence diagram $PD(\mathcal{G}_i, g)$ via the SP vectorization method $\varphi$. We use the first filtration function $f$ to get a finer look at the graph as $\mathcal{G}_1 \subseteq \ldots \subseteq \mathcal{G}_m = \mathcal{G}$. Then, we use the second filtration function $g$ to obtain the persistence diagrams $PD(\mathcal{G}_i, g)$ of these finer pieces. In a way, we look at $\mathcal{G}$ with a two dimensional resolution $\{\mathcal{G}_{ij}\}$ and we keep track of the evolution of topological features in the induced bifiltration $\{\widehat{\mathcal{G}}_{ij}\}$. The main advantage of this technique is that the outputs are fixed size matrices (or arrays) for each dataset which is highly suitable for various machine learning models.

**Order of the Filtration Functions.** We need to note that for given two filtration functions $f, g$, the order is quite important for our algorithm. In particular, let $\mathbf{M}_\varphi(f, g)$ represent the above construction where we first apply $f$ to get filtering $\{\mathcal{G}_i\}_{i=1}^m$, and then we obtain $m$ different $PD(\mathcal{G}_i, g)$. Hence, $\mathbf{M}_\varphi(f, g)$ would be a $m \times k$ matrix. On the other hand, if we apply $g$ first, we would get a filtering $\{\mathcal{G}^j\}_{j=1}^n$. Then, by using sublevel filtration with $g$ for each $\mathcal{G}^j$, we would get $n$ persistence diagrams $PD(\mathcal{G}^j, f)$. Assuming the produced vectors $\vec{\varphi}_j$ for $[\alpha_1, \alpha_m]$ domain has length $k'$, then $\mathbf{M}_\varphi(g, f)$ would give us $n \times k'$ size matrices. In particular, in the first one $\mathbf{M}_\varphi(f, g)$ we use "horizontal" slicing in the bipersistence module, while in the latter $\mathbf{M}_\varphi(g, f)$, we use "vertical" slicing.

For the question *"Which function should be used first"*, the answer is that the function with more important domain information should be used as second function ($g$ in the original construction), as we get much finer information via persistence diagram $PD(\mathcal{G}_i, g)$ for the second function.

**Higher Dimensional Multipersistence.** Similarly, for $d = 3$, let $f, g, h$ be filtration functions and let $\{\mathcal{G}_{ij}\}$ be the bifiltering of the data, e.g., sublevel filtration for two functions $f, g$. Then, again by using the third function $h$, we find $m \cdot n$ persistence diagrams $\{PD(\mathcal{G}_{ij}, h)\}_{i,j=1}^{m,n}$. Similarly, for a given SP vectorization $\varphi$, one obtains a $1D$-vector $\vec{\varphi}(PD(\mathcal{G}_{ij}, g))$ of size $1 \times k$ for each $i, j$. This produces a $3D$-vector (array) $\mathbf{M}_\varphi$ of size $m \times n \times k$ where $\mathbf{M}_\varphi^{ij} = \vec{\varphi}(PD(\mathcal{G}_{ij}, g))$. For $d > 3$, one could follow a similar route.

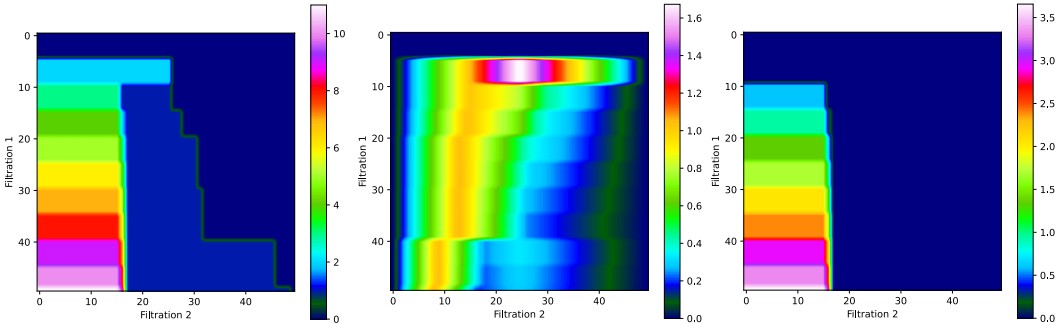

Figure 1: For the same network and the same filtration functions, EMP Betti Summary (left), EMP Silhouette (center), and EMP Entropy Summary (right) can produce highly different topological summaries emphasizing different information in persistence diagrams.

## 4.2 EXAMPLES OF EMP SUMMARIES

Here, we discuss explicit constructions of EMP summaries for most common SP vectorizations. As noted above, the framework is very general and it can be applied to most SP vectorization methods. In all the examples below, we use the following setup: Let $\mathcal{G} = (\mathcal{V}, \mathcal{E})$ be a graph, and let $f, g : \mathcal{V} \to \mathbb{R}$ be two filtration functions with threshold sets $\{\alpha_i\}_{i=1}^m$ and $\{\beta_j\}_{j=1}^n$ respectively. As explained above, we first apply sublevel filtering with $f$ to get a sequence of nested subgraphs, $\mathcal{G}_1 \subseteq \ldots \subseteq \mathcal{G}_m = \mathcal{G}$. Then, for each $\mathcal{G}_i$, we apply sublevel filtration with $g$ to get persistence diagram $PD(\mathcal{G}_i, g)$. Therefore, we will have $m$ PDs. In the examples below, for a given SP vectorization $\varphi$, we explain how to obtain a vector $\vec{\varphi}(PD(\mathcal{G}_i, g))$, and define the corresponding EMP $\mathbf{M}_\varphi$. Note that we skip the homology dimension (subscript $k$ for $PD_k(\mathcal{G})$) in the discussion. In particular, for each dimension $k = 0, 1, \ldots$, we will have one EMP $\mathbf{M}_\varphi(\mathcal{G})$ (a matrix or array) corresponding to $\{\vec{\varphi}(PD_k(\mathcal{G}_i, g))\}$. The most common dimensions are $k = 0$ and $k = 1$ in applications.

**EMP Landscapes.** Persistence Landscapes $\lambda$ are one of the most common SP vectorizations introduced by Bubenik (2015). For a given persistence diagram $PD(\mathcal{G}) = \{(b_i, d_i)\}$, $\lambda$ produces a function $\lambda(\mathcal{G})$ by using generating functions $\Lambda_i$ for each $(b_i, d_i) \in PD(\mathcal{G})$, i.e., $\Lambda_i : [b_i, d_i] \to \mathbb{R}$ is a piecewise linear function obtained by two line segments starting from $(b_i, 0)$ and $(d_i, 0)$ connecting to the same point $\left(\frac{b_i + d_i}{2}, \frac{b_i - d_i}{2}\right)$. Then, the *Persistence Landscape* function $\lambda(\mathcal{G}) : [\epsilon_1, \epsilon_q] \to \mathbb{R}$ for $t \in [\epsilon_1, \epsilon_q]$ is defined as

$$\lambda(\mathcal{G})(t) = \max_i \Lambda_i(t),$$

where $\{\epsilon_k\}_1^q$ are thresholds for the filtration used.

Considering the piecewise linear structure of the function, $\lambda(\mathcal{G})$ is completely determined by its values at $2q - 1$ points, i.e., $\frac{b_i \pm d_i}{2} \in \{\epsilon_1, \epsilon_{1.5}, \epsilon_2, \epsilon_{2.5}, \ldots, \epsilon_q\}$ where $\epsilon_{k.5} = (\epsilon_k + \epsilon_{k+1})/2$. Hence, a vector of size $1 \times (2q - 1)$ whose entries the values of this function would suffice to capture all the information needed, i.e. $\vec{\lambda} = [\lambda(\epsilon_1) \ \lambda(\epsilon_{1.5}) \ \lambda(\epsilon_2) \ \lambda(\epsilon_{2.5}) \ \lambda(\epsilon_3) \ \ldots \ \lambda(\epsilon_q)]$

Considering we have threshold set $\{\beta_j\}_{j=1}^n$ for the second filtration function $g$, $\vec{\lambda}_i = \vec{\lambda}(PD(\mathcal{G}_i, g))$ will be a vector of size $1 \times 2n - 1$. Then, as $\mathbf{M}_\lambda^i = \vec{\lambda}_i$ for each $1 \leq i \leq m$, EMP Landscape $\mathbf{M}_\lambda(\mathcal{G})$ would be a $2D$-vector (matrix) of size $m \times (2n - 1)$.

**EMP Silhouettes.** Silhouette Chazal et al. (2014) is another very popular SP vectorizations method in machine learning applications. The idea is similar to Persistence Landscapes, but this vectorization uses the life span of the topological features more effectively. For $PD(\mathcal{G}) = \{(b_i, d_i)\}_{i=1}^N$, let $\Lambda_i$ be the generating function for $(b_i, d_i)$ as defined in Landscapes above. Then, *Silhouette* function $\psi$ is defined as

$$\psi(\mathcal{G}) = \frac{\sum_{i=1}^N w_i \Lambda_i(t)}{\sum_{i=1}^m w_i}, \ t \in [\epsilon_1, \epsilon_q],$$

where the weight $w_i$ is mostly chosen as the life span $d_i - b_i$, and $\{\epsilon_k\}_{k=1}^q$ represents the thresholds for the filtration used. Again such a Silhouette function $\psi(\mathcal{G})$ produces a $1D$-vector $\vec{\psi}(\mathcal{G})$ of size $1 \times (2q - 1)$ as in Persistence Landscapes case. Similar to the EMP Landscapes, with the threshold set

$\{\beta_j\}_{j=1}^n$ for the second filtration function $g$, $\vec{\psi}_i = \vec{\psi}(PD(\mathcal{G}_i, g)$ will be a vector of size $1 \times 2n - 1$. Then, as $\mathbf{M}_\psi^i = \vec{\psi}_i$ for each $1 \leq i \leq m$, EMP Landscape $\mathbf{M}_\psi(\mathcal{G})$ would be again a $2D$-vector (matrix) of size $m \times (2n - 1)$ (Figure 1).

**EMP Persistence Images.** Next SP vectorization in our list is Persistence Images Adams et al. (2017). Different than the most SP vectorizations, Persistence Images produces $2D$-vectors. The idea is to capture the location of the points in the persistence diagrams with a multivariable function by using the $2D$ Gaussian functions centered at these points. For $PD(\mathcal{G}) = \{(b_i, d_i)\}$, let $\phi_i$ represent a $2D$-Gaussian centered at the point $(b_i, d_i) \in \mathbb{R}^2$. Then, one defines a multivariable function, *Persistence Surface*, $\widetilde{\mu} = \sum_i w_i \phi_i$ where $w_i$ is the weight, mostly a function of the life span $d_i - b_i$. To represent this multivariable function as a $2D$-vector, one defines a $k \times l$ grid (resolution size) on the domain of $\widetilde{\mu}$, i.e., threshold domain of $PD(\mathcal{G})$. Then, one obtains the *Persistence Image*, a $2D$-vector (matrix) $\vec{\mu} = [\mu_{rs}]$ of size $k \times l$ where $\mu_{rs} = \int_{\Delta_{rs}} \widetilde{\mu}(x, y) \, dxdy$ and $\Delta_{rs}$ is the corresponding pixel (rectangle) in the $k \times l$ grid.

This time, the resolution size $k \times l$ is independent of the number of thresholds used in the filtering, the choice of $k$ and $l$ is completely up to the user. Recall that by applying the first function $f$, we have the nested subgraphs $\{\mathcal{G}_i\}_{i=1}^m$. For each $\mathcal{G}_i$, the persistence diagram $PD(\mathcal{G}_i, g)$ obtained by sublevel filtration with $g$ induces a $2D$ vector $\vec{\mu}_i = \vec{\mu}(PD(\mathcal{G}_i, g))$ of size $k \times l$. Then, define EMP Persistence Image as $\mathbf{M}_\mu^i = \vec{\mu}_i$, where $\mathbf{M}_\mu^i$ is the $i^{th}$-floor of the array $\mathbf{M}_\mu$. Hence, $\mathbf{M}_\mu(\mathcal{G})$ would be a $3D$-vector (array) of size $m \times k \times l$ where $m$ is the number of thresholds for the first function $f$ and $k \times l$ is the chosen resolution size for the Persistence Image $\vec{\mu}$.

**EMP Surfaces and EMP Betti Summaries.** Next, we give an important family of SP vectorizations, Persistence Curves Chung & Lawson (2019). This is an umbrella term for several different SP vectorizations, i.e., Betti Curves, Life Entropy, Landscapes, et al. Our EMP framework naturally adapts to all Persistence Curves to produce multidimensional vectorizations. As Persistence Curves produce a single variable function in general, they all can be represented as $1D$-vectors by choosing a suitable mesh size depending on the number of thresholds used. Here, we describe one of the most common Persistence Curves in detail, i.e., Betti Curves. It is straightforward to generalize the construction to other Persistence Curves.

Betti curves are one of the simplest SP vectorization as it gives the count of topological feature at a given threshold interval. In particular, $\beta_k(\Delta)$ is the total count of $k$-dimensional topological feature in the simplicial complex $\Delta$, i.e., $\beta_k(\Delta) = rank(H_k(\Delta))$ (See Figure 2 in the Appendix). In particular, $\beta_k(\mathcal{G}) : [\epsilon_1, \epsilon_{q+1}] \to \mathbb{R}$ is a step function defined as

$$\beta_k(\mathcal{G})(t) = rank(H_k(\widehat{\mathcal{G}}_i))$$

for $t \in [\epsilon_i, \epsilon_{i+1})$, where $\{\epsilon_i\}_1^q$ represents the thresholds for the filtration used. Considering this is a step function where the function is constant for each interval $[\epsilon_i, \epsilon_{i+1})$, it can be perfectly represented by a vector of size $1 \times q$ as $\vec{\beta}(\mathcal{G}) = [\beta(1) \ \beta(2) \ \beta(3) \ \ldots \ \beta(q)]$.

Then, with the threshold set $\{\beta_j\}_{j=1}^n$ for the second filtration function $g$, $\vec{\beta}_i = \vec{\beta}(PD(\mathcal{G}_i, g))$ will be a vector of size $1 \times n$. Then, as $\mathbf{M}_\beta^i = \vec{\beta}_i$ for each $1 \leq i \leq m$, EMP Betti Summary $\mathbf{M}_\beta(\mathcal{G})$ would be a $2D$-vector (matrix) of size $m \times n$ (Figure 1). In particular, each entry $\mathbf{M}_\beta = [m_{ij}]$ is just the Betti number of the corresponding clique complex in the bifiltration $\{\widehat{\mathcal{G}}_{ij}\}$, i.e., $m_{ij} = \beta(\widehat{\mathcal{G}}_{ij})$. This matrix $\mathbf{M}_\beta$ is also called *bigraded Betti numbers* in the literature, and computationally much faster than other vectorizations Lesnick & Wright (2022).

### 4.3 STABILITY OF EMP SUMMARIES

We now show that when the source single parameter vectorization $\varphi$ is stable, then so is its induced EMP vectorization $\mathbf{M}_\varphi$. (We give the details of stability notion in persistence theory and examples of stable SP vectorizations in Appendix C.1.)

Let $\mathcal{G}^+ = (\mathcal{V}^+, \mathcal{E}^+)$ and $\mathcal{G}^- = (\mathcal{V}^-, \mathcal{E}^-)$ be two graphs. Let $\varphi$ be a stable SP vectorization with the stability equation

$$\mathrm{d}(\varphi(\mathcal{G}^+), \varphi(\mathcal{G}^-)) \leq C_\varphi \cdot \mathcal{W}_{p_\varphi}(PD(\mathcal{G}^+), PD(\mathcal{G}^-)) \tag{1}$$

for some $1 \leq p_{\varphi} \leq \infty$. Here, $\varphi(\mathcal{G}^{\pm})$ represent the corresponding vectorizations for $PD(\mathcal{G}^{\pm})$ and $\mathcal{W}_p$ represents Wasserstein-$p$ distance as defined in Appendix C.1.

Now, by taking $d = 2$ for EMP construction, let $f, g : \mathcal{V}^{\pm} \to \mathbb{R}$ be two filtration functions with threshold sets $\{\alpha_i\}_{i=1}^m$ and $\{\beta_j\}_{j=1}^n$ respectively. Then, by defining $\mathcal{V}_i^{\pm} = \{v_r \in \mathcal{V}^{\pm} \mid f(v_r) \leq \alpha_i\}$, their induced subgraphs $\{\mathcal{G}_i^{\pm}\}$ give the filtration $\widehat{\mathcal{G}}_1 \subset \widehat{\mathcal{G}}_2 \subset \ldots \widehat{\mathcal{G}}_m$ as before. For each $1 \leq i \leq m$, we will have persistence diagram $PD(\mathcal{G}_i, g)$ as detailed in Section 4.1. We define the induced matching distance between the multiple persistence diagrams as

$$\mathbf{D}_p(\{PD(\mathcal{G}_i^+)\}, \{PD(\mathcal{G}_i^-)\}) = \sum_{i=1}^m \mathcal{W}_p(PD(\mathcal{G}_i^+, g), PD(\mathcal{G}_i^-, g)). \tag{2}$$

Now, we define the distance between induced EMP Summaries as

$$\mathfrak{D}(\mathbf{M}_{\varphi}(\mathcal{G}^+), \mathbf{M}_{\varphi}(\mathcal{G}^-)) = \sum_{i=1}^m \mathrm{d}(\varphi(\mathcal{G}_i^+), \varphi(\mathcal{G}_i^-)) \tag{3}$$

**Theorem 4.1.** *Let $\varphi$ be a stable SP vectorization. Then, the induced EMP Vectorization $\mathbf{M}_{\varphi}$ is also stable, i.e., with the notation above, there exists $\widehat{C}_{\varphi} > 0$ such that for any pair of graphs $\mathcal{G}^+$ and $\mathcal{G}^-$, we have the following inequality.*

$$\mathfrak{D}(\mathbf{M}_{\varphi}(\mathcal{G}^+), \mathbf{M}_{\varphi}(\mathcal{G}^-)) \leq \widehat{C}_{\varphi} \cdot \mathbf{D}_{p_{\varphi}}(\{PD(\mathcal{G}^+)\}, \{PD(\mathcal{G}^-)\})$$

The proof is given in Appendix C.2.

## 5 EXPERIMENTS

### 5.1 DATASETS

To verify our EMP summaries on graph representation learning, we evaluate EMP summaries with random forest on nine widely used benchmark datasets, i.e., (i) three molecule graphs (Sutherland et al., 2003): BZR_MD, COX2_MD, and DHFR_MD; (ii) two biological graphs (Kriege & Mutzel, 2012; Borgwardt et al., 2005): MUTAG and PROTEINS; and (iii) four social graphs: IMDB-Binary (IMDB-B), IMDB-Multi (IMDB-M), REDDIT-Binary (REDDIT-B), and REDDIT-Multi-5K (REDDIT-M-5K). We also calculate (i) average node-level centrality measures: average degree centrality, average betweenness centrality, average closeness centrality, average clustering centrality, average eigenvector centrality, and average Katz centrality; and (ii) average edge-level centrality measures: average betweenness centrality, average Ricci curvature, and average Ricci positive curvature, and feed these node-/edge-level measures into random forest model. Furthermore, it is worth noting that COX2_MD and DHFR_MD all have three node features, whereas PROTEINS has one node feature. We use all these available features as input to a random forest model. Further details on the datasets and experimental setup are provided in Appendix E.1.

### 5.2 EXPERIMENTAL SETUP

We vectorize our proposed EMP representations and use random forest (RF) for the graph classification task. For RF, we set the number of trees in the forest as 1000, the minimum number of samples as 2, and the function to measure the split quality as Gini impurity. We apply filtrations based on the available information of each dataset, either using their graph structure or their provided node/edge features. Our pool of filtration functions include, but is not limited to: type of atom, closeness, edge-betweenness, weighted degree, katz centrality, ricci curvature, and distance between atoms. In this experiments, first we apply a sublevel filtration on nodes, and secondly a power filtration over edges. To test our EMP framework we use three vectorizations: Betti curves, silhouette functions, and entropy summary functions, thus producing EMP matrix representations which can be embedded in classic and modern machine learning algorithms.

We compare EMP summary with two types of state-of-the-art baselines, covering six graph kernels, one graph neural networks, and one topological method. Graph kernels: (i) comprised of the

Table 1: Classification accuracy (in % $\pm$ standard deviation) of EMP summary on nine benchmark datasets. The best results are in **bold** font and the second best results are marked *underlined*.

| Model | BZR_MD | COX2_MD | DHFR_MD | MUTAG | PROTEINS | IMDB-B | IMDB-M | REDDIT-B | REDDIT-5K |
|---|---|---|---|---|---|---|---|---|---|
| CSM | 77.63±1.29 | OOT | OOT | 87.29±1.25 | OOT | OOT | OOT | OOT | OOT |
| HGK-SP | 60.08±0.88 | 59.92±0.66 | 67.95±0.00 | 80.90±0.48 | 74.53±0.35 | 73.34±0.47 | **51.58±0.42** | OOM | OOM |
| HGK-WL | 52.64±1.20 | 57.15±1.20 | 66.08±1.02 | 75.51±1.34 | 74.53±0.35 | 72.75±1.02 | 50.73±0.63 | OOM | OOM |
| MLG | 51.46±0.61 | 51.15±0.00 | 67.95±0.00 | 78.53±2.25 | **75.55±0.71** | 52.56±0.42 | 34.27±0.33 | OOM | OOM |
| WL | 67.45±1.40 | 60.07±2.22 | 62.56±1.51 | 85.75±1.96 | 73.06±0.47 | 71.15±0.47 | 50.25±0.72 | 77.95±0.60 | 51.63±0.37 |
| WL-OA | 68.19±1.09 | 62.37±2.11 | 64.10±1.70 | 86.10±1.95 | 73.50±0.87 | 74.01±0.66 | 49.95±0.46 | 87.60±0.33 | OOM |
| GNNs | 69.87±1.29 | 66.05±3.16 | 73.11±1.59 | 80.42±2.07 | 70.31±1.93 | 66.53±2.33 | 48.93±0.88 | 89.90±1.90 | **56.10±1.60** |
| FC-V | 75.61±1.13 | **73.41±0.79** | 76.78±0.69 | 87.31±0.66 | 74.54±0.48 | 73.84±0.36 | 46.80±0.37 | 89.41±0.24 | 52.36±0.37 |
| **EMP** | **77.77±0.95** | 70.48±1.38 | **80.50±1.07** | **88.79±0.63** | 72.78±0.54 | **75.34±0.29** | 48.75±0.22 | **91.03±0.22** | 54.41±0.32 |

subgraph matching kernel (CSM) Kriege & Mutzel (2012), (ii) Shortest Path Hash Graph Kernel (HGK-SP) Morris et al. (2016), (iii) Weisfeiler–Lehman Hash Graph Kernel (HGK-WL) Morris et al. (2016), (iv) Multiscale Laplacian Graph Kernel (MLG) Kondor & Pan (2016), (v) Weisfeiler–Lehman (WL) Shervashidze et al. (2011), and (vi) Weisfeiler-Lehman Optimal Assignment (WL-OA) Kriege et al. (2016); graph neural networks: graph neural networks (GNN) Kipf & Welling (2017); topological method: filtration curves (FC-V) O'Bray et al. (2021). For all methods, we report the average accuracy of 10 runs of 10-fold CV along with standard deviation, and running time for computation of each EMP representation. The source code is freely available[1].

## 5.3 EXPERIMENTAL RESULTS

Table 1 shows the results of different methods on nine graph datasets. Out-of-time (OOT) results indicate that a method could not complete the classification task within 12 hours, and OOM means "out-of-memory" (from an allocation of 128 GB RAM). We observe the following phenomena:

- Compared with all baselines, out of 9 benchmark datasets, the proposed EMP summaries achieve the best performance on 5 datasets (BZR_MD, DFHR_MD, MUTAG, IMDB-B, REDDIT-B), and second best in 2 datasets (COX2_MD, REDDIT-5K).

- EMP summary consistently outperforms Filtration Curves on all datasets except COX2_MD and PROTEINS, indicating that multiparameter structure of the EMP summaries can better capture the complex structural properties and local topological information in heterogeneous graphs.

- Moreover, EMP Summaries consistently delivers the competitive results with GNNs in all benchmark datasets. This indicates that EMP summary introduces powerful topological and geometric schemes for node features and graph representation learning.

To further investigate the effectiveness of the EMP vectorization function in graph representation learning, we have conducted ablation studies of various EMP summaries on the benchmark datasets, see Appendix G. Furthermore, Appendix F includes an analysis of computational complexity.

## 6 DISCUSSION

We have proposed a new computationally efficient summary for multidimensional persistence for various forms of data, and include focus-cases on graph machine learning. The new Effective Multidimensional Persistence (EMP) framework provides a practical way to employ the promising multidimensional persistence idea in real-life applications. As such, EMP summaries are highly compatible with machine learning models and can boost the performances of popular single persistent summaries in a unified manner. In a graph classification task, EMP summaries outperform state-of-the-art methods in several benchmark datasets. Furthermore, we have shown that EMP maintain important stability guarantees. Therefore, EMP makes an important step toward bringing the theoretical concepts of multipersistence to the machine learning community and advances the use of persistent homology

---

[1]`https://www.dropbox.com/sh/o9ter2umpus04fi/AABj40YYB0JTFVeENt2qfAKXa?dl=0`

in various settings. In future work, we aim to enhance the EMP framework by using more than one slicing direction in the multipersistence grid and, thus, effectively combining the outputs of multiple slicing directions with deep learning approaches.

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

# Appendix

## A   ADDITIONAL LITERATURE ON GRAPH REPRESENTATION LEARNING

After the success of convolutional neural networks (CNN) on image-based tasks, graph neural networks (GNNs) have emerged as a powerful tool for graph classification and representation learning. Based on the spectral graph theory, Bruna et al. (2014) introduced a graph-based convolution in Fourier domain. However, complexity of this model is very high since all Laplacian eigenvectors are needed. To tackle this problem, ChebNet Defferrard et al. (2016) integrated spectral graph convolution with Chebyshev polynomials. Then, Graph Convolutional Networks (GCNs) of Kipf & Welling (2017) simplified the graph convolution with a localized first-order approximation. More recently, there have been proposed various approaches based on accumulation of the graph information from a wider neighborhood, using diffusion aggregation and random walks. Such higher-order methods include approximate personalized propagation of neural predictions (APPNP) Klicpera et al. (2019), higher-order graph convolutional architectures (MixHop) Abu-El-Haija et al. (2019), multi-scale graph convolution (N-GCN) Abu-El-Haija et al. (2020), and Lévy Flights Graph Convolutional Networks (LFGCN) Chen et al. (2020). In addition to random walks, other recent approaches include GNNs on directed graphs (MotifNet) Monti et al. (2018), graph convolutional networks with attention mechanism (GAT, SPAGAN) Veličković et al. (2018); Yang et al. (2019), and graph Markov neural network (GMNN) Qu et al. (2019). Most recently, Liu et al. Liu et al. (2020) consider utilizing information on the node neighbors' features in GNN, proposing Deep Adaptive Graph Neural Network (DAGNN). However, DAGNN, and other state-of-the-art approaches, does not account for the important information on the shapes of the node neighborhoods.

## B   FURTHER BACKGROUND ON SINGLE PERSISTENT HOMOLOGY

Here, we give further details on single parameter persistent homology. To sum up, PH machinery is a 3-step process. The first step is the *filtration* step, where one can integrate the domain information into the process. The second step is the *persistence diagrams*, where the machinery records the evolution of topological features (birth/death times) in the filtration sequence of the simplicial complexes. The final step is the *vectorization* (fingerprinting), where one can convert these records to a function or vector to be used in suitable ML models.

**i. Constructing Filtrations:** As PH is the machinery to keep track of the evolution of topological features in a sequence, the most important step is inducing this nested sequence of simplicial complexes, $\Delta_0 \subset \Delta_1 \subset \cdots \subset \Delta_m$. This is the key step where one can inject valuable domain information into the PH process by using important domain functions. Two most common methods are *Sublevel/superlevel filtration* and *Vietoris-Rips (VR) filtration*. We already described sublevel filtration in Section 3.1.

*VR filtration* is another common method especially used for point clouds, where coarse geometry of the data set $\mathcal{X}$ play key role (Chazal & Michel, 2021). Let $\mathcal{X} = \{x_1, x_2, ..., x_N\}$ be the given data set. For a given threshold $\epsilon_i$, one forms a Vietoris-Rips complex $\Delta_i$ by adding a $k$-simplex to $X$ for any subset $\{x_{n_0}, x_{n_1}, ..., x_{n_k}\}$, where the pairwise distances are all $d(x_{n_r}, x_{n_s}) < \epsilon_i$. In particular, if a pair of points $x_{n_0}, x_{n_1}$ has distance $< \epsilon_i$, then in the induced simplicial complex $\Delta_i$, we add an edge $e_{n_0 n_1}$ between the corresponding vertices $x_{n_0}$ and $x_{n_1}$. If three such points $x_{n_0}, x_{n_1}, x_{n_2}$ have pairwise distances $< \epsilon_i$, then we fill the triangle $e_{n_0 n_1} \cup e_{n_1 n_2} \cup e_{n_2 n_0}$ with a 2-simplex, and so on. For each $\epsilon_i$, one obtains a simplicial complex $\Delta_i$ by using this procedure. Changing threshold values $\epsilon_1 < \epsilon_2 < \ldots < \epsilon_m$ results in a hierarchical nested sequence of simplicial complexes $\Delta_1 \subset \Delta_2 \subset \cdots \subset \Delta_m$ that is termed *the Vietoris-Rips filtration* of the data set $\mathcal{X}$. Note that VR filtration can also be considered as a sublevel filtration for the distance function to $\mathcal{X}$ from the ambient space $\mathbb{R}^d$, i.e., $f : \mathbb{R}^d \to \mathbb{R}$ with $f(y) = d(y, \mathcal{X})$ where $\mathcal{X} \subset \mathbb{R}^d$.

**ii. Persistence Diagrams:** The second step in PH process is to obtain persistence diagrams (PD) for the filtration $\Delta_0 \subset \Delta_1 \subset \cdots \subset \Delta_m$. As explained in Section 3.1, PDs are collection of 2-tuples, marking the birth and death times of the topological features appearing in the filtration, i.e. $\mathrm{PD}_k(\mathcal{X}) = \{(b_\sigma, d_\sigma) \mid \sigma \in H_k(\Delta_i) \text{ for } b_\sigma \leq i < d_\sigma\}$. Current algorithmic advancements make possible to efficiently compute persistence diagrams via modern software libraries (Otter et al., 2017).

**iii. Vectorizations:** While PH extracts hidden shape patterns from data as persistence diagrams (PD), PDs, a collection of points in $\mathbb{R}^2$ by itself, are not very practical for statistical and ML purposes. Instead, the common techniques are faithfully representing PDs as kernels (Kriege et al., 2020) or vectorizations (Hensel et al., 2021). One can consider this step as converting PDs into a useful format to be used in the ML process as fingerprints of the dataset. This provides a practical way to use the outputs of PH in real-life applications. *Single Persistence Vectorizations* transform obtained PH information (PDs) into a function or a feature vector form which is much more suitable for ML tools than PDs. Common single persistence (SP) vectorization methods are Persistence Images (Adams et al., 2017), Persistence Landscapes (Bubenik, 2015), Silhouettes (Chazal et al., 2014), Betti Curves and various Persistence Curves (Chung & Lawson, 2019). These vectorizations define a single variable or multivariable function out of PDs, which can be used as fixed-size $1D$ or $2D$ vectors in applications, i.e., $1 \times n$ vectors or $m \times n$ vectors. For example, a Betti curve for a PD with $n$ thresholds can also be expressed as $1 \times n$ size vectors. Similarly, Persistence Images is an example of $2D$ vectors with the chosen resolution (grid) size. See the examples given in Section 4.2 for further details.

## C  STABILITY

### C.1  STABILITY OF SINGLE PERSISTENCE SUMMARIES

For a given PD vectorization, the stability is one of the most important properties for statistical purposes. Intuitively, stability question is whether a small perturbation in PD cause a big change in the vectorization or not. To make this question meaningful, one needs to define what "small" and "big" means in this context. Therefore, we need to define distance notion, i.e., metric in the space of persistence diagrams. The most common such metric is called *Wasserstein distance* (or matching distance) which is defined as follows. Let $PD(\mathcal{X}^+)$ and $PD(\mathcal{X}^-)$ be persistence diagrams two datasets $\mathcal{X}^+$ and $\mathcal{X}^-$ (We omit the dimensions in PDs). Let $PD(\mathcal{X}^+) = \{q_j^+\} \cup \Delta^+$ and $PD(\mathcal{X}^-) = \{q_l^-\} \cup \Delta^-$ where $\Delta^{\pm}$ represents the diagonal (representing trivial cycles) with infinite multiplicity. Here, $q_j^+ = (b_j^+, d_j^+) \in PD(\mathcal{X}^+)$ represents the birth and death times of a hole $\sigma_j$ in $\mathcal{X}^+$. Let $\phi : PD(\mathcal{X}^+) \to PD(\mathcal{X}^-)$ represent a bijection (matching). With the existence of the diagonal $\Delta^{\pm}$ in both sides, we make sure the existence of these bijections even if the cardinalities $|\{q_j^+\}|$ and $|\{q_l^-\}|$ are different. Then, the $p^{th}$ Wasserstein distance $\mathcal{W}_p$ defined as

$$\mathcal{W}_p(PD(\mathcal{X}^+), PD(\mathcal{X}^-)) = \min_{\phi}\left(\sum_j \|q_j^+ - \phi(q_j^+)\|_{\infty}^p\right)^{\frac{1}{p}}, \quad p \in \mathbb{Z}^+.$$

Then, a vectorization (function) $\varphi(PD(\mathcal{X}))$ is called *stable* if $\mathrm{d}(\varphi^+, \varphi^-) \leq C \cdot \mathcal{W}_p(PD(\mathcal{X}^+), PD(\mathcal{X}^-))$ where $\varphi^{\pm} = \varphi(PD(\mathcal{X}^{\pm}))$ and $\mathrm{d}(.,.)$ is a suitable metric on the space of vectorizations used. Here, the constant $C > 0$ is independent of $\mathcal{X}^{\pm}$. This stability inequality interprets as the changes in the vectorizations are bounded by the changes in PDs. Two nearby persistence diagrams are represented by nearby vectorizations. If a given vectorization $\varphi$ satisfies such a stability inequality for some d and $\mathcal{W}_p$, we call $\varphi$ a *stable vectorization* Atienza et al. (2020). Persistence Landscapes Bubenik (2015), Persistence Images Adams et al. (2017), Stabilized Betti Curves Johnson & Jung (2021) and several Persistence curves Chung & Lawson (2019) are among well-known examples of stable vectorizations.

### C.2  PROOF OF THEOREM 4.1: STABILITY OF EMP SUMMARIES

*Proof.* As $\varphi$ is a stable SP vectorization, for any $1 \leq i \leq m$, we have $\mathrm{d}(\varphi(\mathcal{G}_i^+), \varphi(\mathcal{G}_i^-)) \leq C_{\varphi} \cdot \mathcal{W}_{p_{\varphi}}(PD(\mathcal{G}_i^+), PD(\mathcal{G}_i^-))$ for some $C_{\varphi} > 0$ by Equation (1), where $\mathcal{W}_{p_{\varphi}}$ is Wasserstein-$p$

distance. Notice that the constant $C_\varphi > 0$ is independent of $i$. Hence,

$$
\begin{aligned}
\mathfrak{D}(\mathbf{M}_\varphi(\mathcal{G}^+), \mathbf{M}_\varphi(\mathcal{G}^-)) \quad &= \quad \sum_{i=1}^{m} \mathrm{d}(\varphi(\mathcal{G}_i^+), \varphi(\mathcal{G}_i^-)) \\
&\leq \quad \sum_{i=1}^{m} C_\varphi \cdot \mathcal{W}_{p_\varphi}(PD(\mathcal{G}_i^+), PD(\mathcal{G}_i^-)) \\
&= \quad C_\varphi \sum_{i=1}^{m} \mathcal{W}_{p_\varphi}(PD(\mathcal{G}_i^+), PD(\mathcal{G}_i^-)) \\
&= \quad C_\varphi \cdot \mathbf{D}_{p_\varphi}(\{PD(\mathcal{G}_i^+)\}, \{PD(\mathcal{G}_i^-)\})
\end{aligned}
$$

where the first and last equalities are due to Equation (2) and Equation (3), while the inequality follows from Equation (1) which is true for any $i$. This concludes the proof of the theorem. $\qquad\square$

## D    EMP FRAMEWORK

### D.1    EMP FOR OTHER TYPES OF DATA

So far, to keep the exposition simple, we described our construction in the graph setup. However, our framework is suitable for various types of data. Let $\mathcal{X}$ be a an image data or a point cloud. Let $f : \mathcal{X} \to \mathbb{R}$ and $g : \mathcal{X} \to \mathbb{R}$ be two filtration functions on $\mathcal{X}$. For example, it can be grayscale function for image data, or density function on point cloud data.

Let $f : \mathcal{X} \to \mathbb{R}$ be the filtration function with threshold set $\{\alpha_i\}_1^m$. Let $\mathcal{X}_i = f^{-1}((-\infty, \alpha_i])$. Then, we get a filtering of $\mathcal{X}$ as nested subspaces $\mathcal{X}_1 \subset \mathcal{X}_2 \subset \cdots \subset \mathcal{X}_m = \mathcal{X}$. By using the second filtration function, we obtain finer filtrations for each subspace $\mathcal{X}_i$ where $1 \leq i \leq m$. In particular, fix $1 \leq i_0 \leq m$ and let $\{\beta_j\}_{j=1}^n$ be the threshold set for the second filtration function $g$. Then, by restricting $g$ to $\mathcal{X}_{i_0}$, we get a filtration function on $X_{i_0}$, i.e., $g : \mathcal{X}_{i_0} \to \mathbb{R}$ which produces filtering $\mathcal{X}_{i_0 1} \subset \mathcal{X}_{i_0 2} \subset \cdots \subset \mathcal{X}_{i_0 n} = \mathcal{X}_{i_0}$. By inducing a simplicial complex $\widehat{\mathcal{X}}_{i_0 j}$ for each $\mathcal{X}_{i_0 j}$, we get a filtration $\widehat{\mathcal{X}}_{i_0 1} \subset \widehat{\mathcal{X}}_{i_0 2} \subset \cdots \subset \widehat{\mathcal{X}}_{i_0 n} = \widehat{\mathcal{X}}_{i_0}$. This filtration results in a persistence diagram (PD) $PD(\mathcal{X}_{i_0}, g)$. For each $1 \leq i \leq m$, we obtain $PD(\mathcal{X}_i, g)$. Note that after getting $\{\mathcal{X}_i\}_{i=1}^m$ via $f$, instead of using second filtration function $g$, one can apply power filtration or Vietoris-Rips construction based on distance for each $X_{i_0}$ in order to get a different filtration $\widehat{\mathcal{X}}_{i_0 1} \subset \widehat{\mathcal{X}}_{i_0 2} \subset \cdots \subset \widehat{\mathcal{X}}_{i_0 n} = \widehat{\mathcal{X}}_{i_0}$.

By using $m$ PDs, we follow a similar route to define our EMP summaries. Let $\varphi$ be a single persistence vectorization. By applying the chosen SP vectorization $\varphi$ to each PD, we obtain a function $\varphi_i = \varphi(PD(\mathcal{X}_i, g))$ on the threshold domain $[\beta_1, \beta_n]$, which can be expresses as a $1D$ (or $2D$) vector in most cases (Section 4.2). Let $\vec{\varphi}_i$ be the corresponding $1 \times k$ vector for the function $\varphi_i$. Define the corresponding EMP $\mathbf{M}_\varphi$ as $\mathbf{M}_\varphi^i = \vec{\varphi}_i$ where $\mathbf{M}_\varphi^i$ is the $i^{th}$ row of $\mathbf{M}_\varphi$. In particular, $\mathbf{M}_\varphi$ is a $2D$-vector (a matrix) of size $m \times k$ where $m$ is the number of thresholds for the first filtration function $f$, and $k$ is the length of the vector $\vec{\varphi}$.

### D.2    EMP WITH OTHER FILTRATIONS

**Weight filtration**  For a given weighted graph $\mathcal{G} = (\mathcal{V}, \mathcal{E}, \mathcal{W})$, it is common to use edge weights $\mathcal{W} = \{\omega_{rs} \in \mathbb{R}^+ \mid \epsilon_{rs} \in \mathcal{E}\}$ to describe filtration. By choosing the threshold set similarly $\mathcal{I} = \{\alpha_i\}_1^m$ with $\alpha_1 = \min\{\omega_{rs} \in \mathcal{W}\} < \alpha_2 < \ldots < \alpha_m = \max\{\omega_{rs} \in \mathcal{W}\}$. For $\alpha_i \in \mathcal{I}$, let $\mathcal{E}_i = \{e_{rs} \in \mathcal{V} \mid \omega_{rs} \leq \alpha_i\}$. Let $\mathcal{G}^i$ be a subgraph of $\mathcal{G}$ induced by $\mathcal{V}_i$. This induces a nested sequence of subgraphs $\mathcal{G}_1 \subset \mathcal{G}_2 \subset \cdots \subset \mathcal{G}_m = \mathcal{G}$ (See top row in Figure 2).

In the case of weighted graphs, one can apply the EMP framework just by replacing the first filtering (via $f$) with weight filtering. In particular, let $g : \mathcal{V} \to \mathbb{R}$ be a filtration function with threshold set $\{\beta_j\}_{j=1}^n$. Then, one can first apply weight filtering to get $\mathcal{G}_1 \subset \cdots \subset \mathcal{G}_m = \mathcal{G}$ as above, and then apply $f$ to each $\mathcal{G}_i$ to get a bilfiltration $\{\mathcal{G}_{ij}\}$ ($m \times n$ resolution). One gets $m$ PDs as $PD(\mathcal{G}_i, g)$ and induce the corresponding $\mathbf{M}_\varphi$. Alternatively, one can change the order by applying $g$ first, and get a different filtering $\mathcal{G}_1 \subset \mathcal{G}_2 \subset \cdots \subset \mathcal{G}_n = \mathcal{G}$ induced by $g$. Then, apply to edge weight filtration

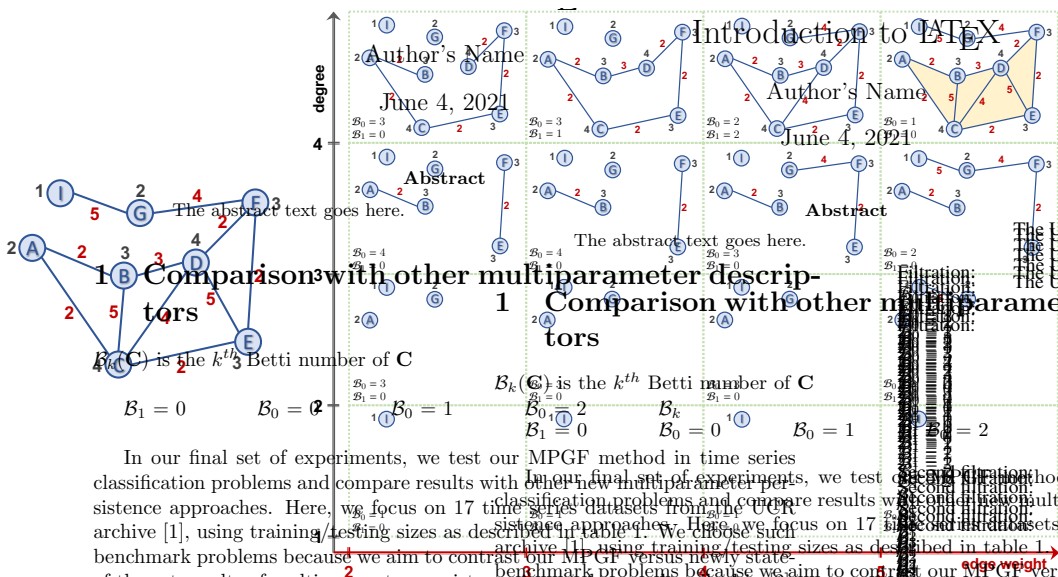

Figure 2: Multidimensional persistence on a graph network (original graph: left). Black numbers denote the degree values of each node whilst red numbers show the edge weights of the network. Hence, shape properties are computed on two filtration functions (i.e., degree and edge weight). While each row filters by degree, each column filters the corresponding subgraph using its edge weights. For each cell, lower left corners represent the corresponding threshold values. For each cell, $\mathcal{B}_0$ and $\mathcal{B}_1$ represent the corresponding Betti numbers.

to any $\mathcal{G}_j$, one gets a bifiltration $\{\widehat{\mathcal{G}}_{ji}\}$ ($n \times m$ resolution) this time. As a result, one gets $n$ PDs as $PD(\mathcal{G}_i, \omega)$ and induce the corresponding $\mathbf{M}_\varphi$. The difference is that in the first case (first apply weights, then $g$), the filtration function plays more important role as $\mathbf{M}_\varphi$ uses $PD(\mathcal{G}_i, g)$ while in the second case (first apply $g$, then apply weights) weights have more important role as $\mathbf{M}_\varphi$ is induced by $PD(\mathcal{G}_j, \omega)$. Note also that there is a very different filtration method for weighted graphs by applying the following the following VR-complexes method.

**Power (Vietoris-Rips) Filtration** There is a highly different filtration technique using distances between the data points in the dataset. The technique is called *power filtration* for unweighted graphs Aktas et al. (2019), while it is called *Vietoris-Rips filtration* for other types of data Edelsbrunner & Harer (2010). The idea is for a point cloud $\mathcal{X} = \{x_1, x_2, \ldots, x_N\}$, one uses the pairwise distances $d(x_r, x_s)$ to construct the simplicial complexes in the filtration. In particular, for a threshold set $\epsilon_1 < \epsilon_2 < \cdots < \epsilon_n = diam(\mathcal{X})$, one forms a Vietoris-Rips complex $\Delta_j$ by adding a $k$-simplex to $\mathcal{X}$ for any subset $\{x_{r_0}, x_{r_1}, \ldots, x_{r_k}\}$, where the pairwise distances are all $< \epsilon_j$. If a pair of points $x_{r_1}, x_{r_2}$ has distance $< \epsilon_j$, then in the induced simplicial complex $\Delta_j$, we add an edge between the corresponding vertices $x_r$ and $x_s$. If three such points $x_{r_1}, x_{r_2}, x_{r_3}$ have pairwise distances $< \epsilon_j$, then we fill the triangle $e_{r_1 r_2} \cup e_{r_2 r_3} \cup e_{r_3 r_1}$ with a 2-simplex, and so on. This procedure induces in a hierarchical nested sequence of simplicial complexes $\Delta_1 \subset \Delta_2 \subset \ldots \subset \Delta_m$ that is termed *Vietoris-Rips filtration* induced by the point cloud $\mathcal{X}$. For unweighted graphs, one takes the vertex set $\mathcal{V}$ as the point cloud, and defines the distances $d(v_i, v_j)$ as the shortest distance in the graph where each edge has length 1. For weighted graphs, one can do the same by defining edge lengths induced by the weights.

One can adapt Vietoris-Rips filtrations to our EMP setting as follows. Start with a filtration function $f : \mathcal{X} \to \mathbb{R}$ with threshold set $\{\alpha_i\}_1^m$ and obtain $\mathcal{X}_1 \subset \mathcal{X}_2 \subset \cdots \subset \mathcal{X}_m = \mathcal{X}$ where $\mathcal{X}_i = f^{-1}((-\infty, \alpha_i])$. Then, apply Vietoris-Rips filtration to each $\mathcal{X}_{i_0}$ for threshold set $\{\epsilon_j\}_{j=1}^n$ which produces a filtration $\widehat{\mathcal{X}}_{i_0 1} \subset \widehat{\mathcal{X}}_{i_0 2} \subset \cdots \subset \widehat{\mathcal{X}}_{i_0 n}$ where $\widehat{\mathcal{X}}_{i_0 j}$ is the Vietoris-Rips complex of $\mathcal{X}_{i_0}$ for threshold $\epsilon_j$. Construct $PD(\mathcal{X}_i, VR)$ of these filtrations for each $1 \leq i \leq m$. The following steps are same Section 4.2. For a given SP vectorization $\varphi$, let $\vec{\varphi}_i$ be the corresponding $1 \times k$ vector induced by $\varphi(PD(\mathcal{X}_i, VR))$ with domain $[\epsilon_1, \epsilon_n]$. Then, define EMP $\mathbf{M}_\varphi$ as $\mathbf{M}_\varphi^i = \vec{\varphi}_i$ where $\mathbf{M}_\varphi^i$ is the $i^{th}$ row of $\mathbf{M}_\varphi$. Again, $\mathbf{M}_\varphi$ is a $2D$-vector (a matrix) of size $m \times k$ where $m$ is the number of thresholds for the filtration function $f$, and $k$ is the length of the vector $\vec{\varphi}$.

## D.3  MULTIPARAMETER PERSISTENCE THEORY

Multipersistence theory is under intense research because of its promise to significantly improve the performance and robustness properties of single persistence theory. While single persistence theory obtains the topological fingerprint of single filtration, a multidimensional filtration with more than one parameter should deliver a much finer summary of the data to be used with ML models. However, because of the technical issues in the theory, multipersistence has not reached to its potential yet and remains largely unexplored by the ML community. Here, we provide a short summary of these technical issues. For further details, Botnan & Lesnick (2022) gives a nice outline of current state of the theory and major obstacles.

In single persistence, the threshold space $\{\alpha_i\}$ being a subset of $\mathbb{R}$, is totally ordered, i.e., birth time < death time for any topological feature appearing in the filtration sequence $\{\Delta_i\}$. By using this property, it was shown that "barcode decomposition" is well-defined in single persistence theory in 1950s [Krull-Schmidt-Azumaya Theorem Botnan & Lesnick (2022)–Theorem 4.2]. This decomposition makes the persistence module $M = \{H_k(\Delta_i)\}_{i=1}^{N}$ uniquely decomposable into barcodes. This barcode decomposition is exactly what we call a *Persistence Diagram*.

However, when one goes to higher dimensions, i.e. $d = 2$, then the threshold set $\{(\alpha_i, \beta_j)\}$ is no longer totally ordered, but becomes partially ordered (Poset). In other words, some indices have ordering relation $(1, 2) < (4, 7)$, while some do not, e.g., (2,3) vs. (1,5). Hence, if one has a multipersistence grid $\{\Delta_{ij}\}$, we no longer can talk about birth time or death time as there is no ordering anymore. Furthermore, Krull-Schmidt-Azumaya Theorem is no longer true for upper dimensions Botnan & Lesnick (2022)–Section 4.2. Hence, for general multipersistence modules barcode decomposition is not possible, and the direct generalization of single persistence to multipersistence fails. On the other hand, even if the multipersistence module has a good barcode decomposition, because of partial ordering, representing these barcodes faithfully is another major problem. Multipersistence modules are an important subject in commutative algebra, where one can find the details of the topic in Eisenbud (2013).

While complete generalization is out of reach for now, several attempts have been tried to utilize MP idea (Lesnick, 2015). One of the first such novel idea came from Lesnick & Wright (2015) where they suggest to use one dimensional slices in the MP grid, and to get the signature of the most dominant features. Later, Carrière & Blumberg (2020) combined several slicing directions (vineyards) and obtained a vectorization by summarizing several persistence diagrams (PDs) in these directions. Slicing techniques use the persistence diagrams of predetermined one-dimensional slices in the multipersistence grid, and then combine (compress) them as one dimensional output Botnan & Lesnick (2022). In that respect, one major issue is that the topological summary highly depends on the predetermined slicing directions in this approach, and how to decide this direction. The other problem is the loss of information when compressing the information in various persistence diagrams.

As explained above, MP approach has still technical problems to reach its full potential, and there are several attempts to utilize this idea. In this paper, we do not claim to solve theoretical problems of multipersistence homology, but offer a novel, highly practical multidimensional topological summary by advancing the existing methods. We use the grid directions in the multipersistence module as natural slicing directions and produce mutidimensional topological summary of the data. As a result, these multidimensional topological fingerprints are capable of capturing very fine topological information hidden in the data. Furthermore, in the case the data provides more than two very important filtration functions, our framework easily accommodates these functions and induces corresponding substructures. Then, our EMP model capture the evolving topological patterns of these substructures and summarize them in matrices and arrays which are highly practical output format to be used with various ML models.

Our model is highly different from previous work mostly because of its practicality and computational efficiency. Among these, the closest method to ours is Carrière & Blumberg (2020) which employs slicing techniques a different way. Like us, they have predetermined slicing options (vineyards), and they compute the single persistence diagrams on these slices and combine them in a unique way by using weight functions induced from lifespans of the topological features in this collection of persistence diagram. In our approach, we use only horizontal slices and do not compress the information. First, choosing horizontal slices computationally very feasible to obtain persistence diagrams. Second, we offer a variety of options on how to vectorize these persistence diagrams.

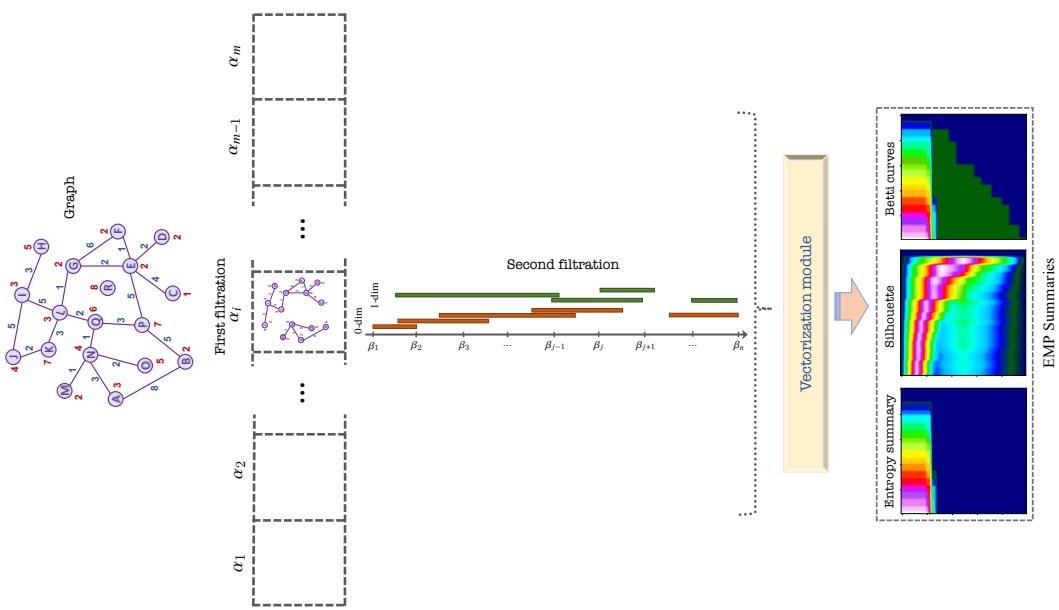

Figure 3: Illustration of the EMP framework for networks. Using the pair of filtration functions $f$, $g$ we define non-decreasing thresholds $\{\alpha_i\}_1^m$ and $\{\beta_j\}_1^n$, respectively, based on node features, red, and edge features, blue. Both, filtrations and vectorizations run in parallel to better use computational resources and produce EMP representations in a timely manner.

Hence, depending on the dataset, one can use vectorization methods which emphasize long barcodes (Silhouette with $p > 1$, Entropy, Persistence Image) or the ones which considers all signals equally (Betti). Our experiments proved that these variety of options can be quite useful as some EMP vectorizations give much better result than others in different datasets (Appendix G).

## E    FURTHER DETAILS ON EXPERIMENTS

### E.1    BENCHMARK DATASETS AND EXPERIMENTAL SETUP

In our experiments, we use 11 benchmark datasets for graph classification tasks (see Table 2). We have run our models for graph classification tasks on an 8-core DO droplet machine with Intel Xeon Scalable processors at a base frequency of 2.5 Ghz. Table 2 summarize the statistics of the datasets in our experiments.

Table 2: Summary statistics of the datasets.

| Dataset | # Graphs | Avg. $|\mathcal{V}|$ | Avg. $|\mathcal{E}|$ | # Class | Node Attr. (Dim.) | Edge Attr. (Dim.) |
|---|---|---|---|---|---|---|
| BZR_MD | 306 | 21.30 | 225.06 | 2 | 3 | - |
| COX2_MD | 303 | 26.28 | 335.12 | 2 | 3 | - |
| DHFR_MD | 393 | 23.87 | 283.02 | 2 | - | 1 |
| MUTAG | 188 | 17.93 | 19.79 | 2 | - | - |
| PROTEINS | 1113 | 39.06 | 72.82 | 2 | 1 | - |
| IMDB-B | 1000 | 19.77 | 96.53 | 2 | - | - |
| IMDB-M | 1500 | 13.00 | 65.94 | 3 | - | - |
| REDDIT-B | 2000 | 429.63 | 497.75 | 2 | - | - |
| REDDIT-M-5K | 4999 | 508.82 | 594.87 | 2 | - | - |

The resolution of vectorization is the most significant parameter, which may impact the computational performance and results. As such, we use a fixed resolution to get consistent results in all experiments and consider time constraints on server usage. We use resolution size of $50\times50$ for each summary

function, and the standard parameters set by the Gudhi library in Python [2]. The order of landscape summary function is set to 1 (max), whilst the power of weights is set to 1 for silhouette summaries.

Results in Table 1 comes from computing MP using filtration functions as follows. BZR_MD, COX2_MD, and DHFR_MD use weighted node-degree and edge-power filtrations. PROTEINS uses node-closeness and edge-betweenness power filtrations. MUTAG, IMDB-BINARY, and IMDB-MULTI use node-katz centrality and edge-ricci curvature power filtrations. REDDIT-BINARY, and REDDIT-MULTI-5K use node-katz centrality and edge-ricci curvature filtrations. We performed an empirical analysis to select previous filter functions for each graph network. We use three types of vectorizations: Betti curves, silhouette functions and entropy summary functions. For each case we compute both 0-dim and 1-dim MP topological features.

## F    COMPUTATIONAL COMPLEXITY

Computational complexity (CC) of persistence diagram $\text{PD}_k(\Delta)$ is $\mathcal{O}(\mathcal{N}^3)$, where $\mathcal{N}$ is the number of $k$-simplices in $\Delta$ (Otter et al., 2017). CC of EMP summary $\mathbf{M}_\varphi^d$ depends on the vectorization $\varphi$ used and the number $d$ of the filtration functions one uses. If $r$ is the resolution size of the multipersistence grid, then one needs $r^{(d-1)}$ single persistence diagrams to obtain $\mathbf{M}_\varphi^d$. Therefore, $\text{CC}(\mathbf{M}_\varphi^d) = \mathcal{O}(r^{(d-1)} \cdot \mathcal{N}^3 \cdot C_\varphi(m))$ where $C_\varphi(m)$ is CC for $\varphi$ and $m$ is the number of barcodes in $\text{PD}_k$, e.g., if $\varphi$ is persistence landscape, then $C_\varphi(m) = m^2$ and hence CC for EMP Landscape with $d = 2$ is $\mathcal{O}(r \cdot \mathcal{N}^3 \cdot m^2)$. In practice, $r$ is a constant and $m$ is small compared to $\mathcal{N}$, hence the complexity is again reduced to $\mathcal{O}(\mathcal{N}^3)$. On the other hand, as Betti numbers do not need $\text{PD}_k$ to be computed, it is possible to obtain much faster algorithms for EMP Betti Summary (Edelsbrunner & Parsa, 2014). Recently, Lesnick & Wright (2022) introduced a quite fast algorithm for EMP Betti summaries with $\mathcal{O}(\mathcal{M}^3)$ time where $\mathcal{M}$ is the rank of the multipersistence module with minimal representation.

## G    ABLATION STUDY

### G.1    DIFFERENT TYPES OF EMP SUMMARIES

Furthermore, to evaluate the performance of different types of EMP summaries (i.e., EMP Silhouette (EMP-S), EMP Entropy (EMP-E), and EMP Betti (EMP-B) with different dimensions (i.e., $H_0$: 0-dimensional topology, and $H_1$: 1-dimensional topology) and graph-based features (i.e., $f_\mathcal{V}$: node features, and $f_\mathcal{E}$: edge features; see more details in Section 5.2 and Appendix E.1). We include comparative statistics using all combinations of input variables, i.e. ablation study, for 3 datasets in our analysis: BZR-MD (Table 3), DHFR-MD (Table 4), and REDDIT-BINARY (Table 5).

Each cell shows the classification accuracy (in % $\pm$ standard deviation) when using different combinations of variables. The best accuracy result is highlighted using bold font. A row/column named as 'none' means that either graph-based features, or EMP summaries where not used to compute the experiments. As such, the top row shows all the accuracy results without using any EMP topological features. The first/left column shows the results only using EMP topological features. The cell in the bottom right corner, of each EMP group, contains the accuracy results using all graph-extracted/EMP features available.

We can observe that: (i) best results contains 1-dimensional EMP summaries, demonstrating the necessity of capturing higher-order structures (e.g., triangles/cycles), (ii) the choice of the EMP summary can significantly affect the performance, and (iii) the addition of EMP topological features generally improves accuracy of versions without EMP summaries, thus, demonstrating the importance of modeling global and topological graph structures.

### G.2    ORDER OF FILTRATION

With our novel approach, especially sublevel/superlevel filtration, we give a computationally very efficient way to extract information of substructures of the data induced by multiple functions. Since

---

[2]https://gudhi.inria.fr/python/latest/

Table 3: Ablation Study on BZR_MD dataset.

|  | none | $f_{\mathcal{V}}$ | $f_{\mathcal{E}}$ | $f_{\mathcal{V}} + f_{\mathcal{E}}$ |
|---|---|---|---|---|
| none | - | $67.124 \pm 1.959$ | $58.714 \pm 0.520$ | $67.642 \pm 1.724$ |
| EMP-B $H_0$ | $67.870 \pm 1.677$ | $68.499 \pm 1.962$ | $67.514 \pm 1.868$ | $68.037 \pm 2.035$ |
| EMP-B $H_1$ | $69.205 \pm 1.239$ | $69.157 \pm 1.220$ | $68.935 \pm 0.818$ | $69.223 \pm 1.058$ |
| EMP-B $H_0 + H_1$ | $68.100 \pm 1.171$ | $68.492 \pm 1.245$ | $68.335 \pm 1.445$ | $69.077 \pm 1.470$ |
| EMP-E $H_0$ | $65.190 \pm 1.484$ | $64.930 \pm 2.045$ | $64.962 \pm 1.682$ | $65.288 \pm 1.754$ |
| EMP-E $H_1$ | $77.469 \pm 1.144$ | $77.438 \pm 0.727$ | $77.441 \pm 0.830$ | $\mathbf{77.766 \pm 0.952}$ |
| EMP-E $H_0 + H_1$ | $73.023 \pm 0.818$ | $72.963 \pm 1.014$ | $73.057 \pm 1.284$ | $73.324 \pm 1.086$ |
| EMP-S $H_0$ | $68.135 \pm 1.294$ | $68.525 \pm 0.865$ | $68.426 \pm 1.235$ | $68.232 \pm 0.969$ |
| EMP-S $H_1$ | $71.068 \pm 1.417$ | $71.071 \pm 1.167$ | $70.906 \pm 1.416$ | $70.743 \pm 1.275$ |
| EMP-S $H_0 + H_1$ | $72.282 \pm 1.012$ | $72.246 \pm 0.883$ | $72.682 \pm 0.913$ | $72.904 \pm 0.901$ |

Table 4: Ablation Study on DHFR_MD dataset.

|  | none | $f_{\mathcal{V}}$ | $f_{\mathcal{E}}$ | $f_{\mathcal{V}} + f_{\mathcal{E}}$ |
|---|---|---|---|---|
| none | - | $68.333 \pm 0.874$ | $60.707 \pm 1.455$ | $68.058 \pm 1.159$ |
| EMP-B $H_0$ | $71.872 \pm 0.804$ | $72.097 \pm 0.876$ | $71.794 \pm 0.541$ | $72.121 \pm 0.943$ |
| EMP-B $H_1$ | $66.638 \pm 0.690$ | $68.190 \pm 0.933$ | $67.072 \pm 0.705$ | $68.774 \pm 1.028$ |
| EMP-B $H_0 + H_1$ | $73.959 \pm 0.746$ | $74.087 \pm 0.820$ | $74.085 \pm 0.671$ | $74.213 \pm 0.905$ |
| EMP-E $H_0$ | $74.906 \pm 0.968$ | $74.550 \pm 0.867$ | $74.521 \pm 1.066$ | $74.523 \pm 1.027$ |
| EMP-E $H_1$ | $66.218 \pm 1.499$ | $66.424 \pm 1.304$ | $66.067 \pm 1.152$ | $66.397 \pm 0.899$ |
| EMP-E $H_0 + H_1$ | $74.947 \pm 0.663$ | $74.849 \pm 0.819$ | $74.667 \pm 0.922$ | $74.796 \pm 0.960$ |
| EMP-S $H_0$ | $78.434 \pm 0.344$ | $78.028 \pm 0.597$ | $78.537 \pm 0.677$ | $78.309 \pm 0.681$ |
| EMP-S $H_1$ | $75.053 \pm 0.909$ | $74.953 \pm 0.930$ | $75.056 \pm 0.785$ | $75.259 \pm 1.051$ |
| EMP-S $H_0 + H_1$ | $80.174 \pm 1.081$ | $\mathbf{80.503 \pm 1.066}$ | $80.174 \pm 0.864$ | $80.126 \pm 0.939$ |

Table 5: Ablation Study on REDDIT-B dataset.

|  | none | $f_{\mathcal{V}}$ | $f_{\mathcal{E}}$ | $f_{\mathcal{V}} + f_{\mathcal{E}}$ |
|---|---|---|---|---|
| none | - | $89.680 \pm 0.300$ | $79.590 \pm 0.394$ | $90.125 \pm 0.277$ |
| EMP-B $H_0$ | $88.925 \pm 0.168$ | $90.400 \pm 0.145$ | $89.145 \pm 0.193$ | $90.520 \pm 0.204$ |
| EMP-B $H_1$ | $80.715 \pm 0.235$ | $87.090 \pm 0.217$ | $84.620 \pm 0.160$ | $88.225 \pm 0.280$ |
| EMP-B $H_0 + H_1$ | $89.970 \pm 0.309$ | $90.945 \pm 0.184$ | $90.025 \pm 0.234$ | $\mathbf{91.025 \pm 0.221}$ |
| EMP-E $H_0$ | $88.140 \pm 0.265$ | $89.875 \pm 0.185$ | $88.400 \pm 0.210$ | $89.945 \pm 0.177$ |
| EMP-E $H_1$ | $79.850 \pm 0.365$ | $87.670 \pm 0.303$ | $84.600 \pm 0.214$ | $88.640 \pm 0.156$ |
| EMP-E $H_0 + H_1$ | $89.285 \pm 0.234$ | $90.010 \pm 0.234$ | $89.240 \pm 0.202$ | $90.045 \pm 0.221$ |
| EMP-S $H_0$ | $86.150 \pm 0.261$ | $87.925 \pm 0.162$ | $86.940 \pm 0.206$ | $88.415 \pm 0.241$ |
| EMP-S $H_1$ | $77.580 \pm 0.186$ | $85.435 \pm 0.332$ | $81.580 \pm 0.222$ | $86.960 \pm 0.346$ |
| EMP-S $H_0 + H_1$ | $86.790 \pm 0.385$ | $88.250 \pm 0.266$ | $87.410 \pm 0.315$ | $88.590 \pm 0.377$ |

we only use horizontal slices, the first function is only used for finer filtration, and the second function gives the persistence diagrams. This makes our method a-symmetric (the order is important). Hence, one can change the order and get different performance results for the model. This asymmetry enriches our method as one can combine both feature vectors obtained by different order as they do not contain the same information about the multipersistence grid. To observe the effect of changing the order of the filtering functions, we run experiments on two benchmark datasets, BZR_MD and

REDDIT-BINARY. Our experiments show that in some datasets, the order can be highly important, while in others, it has negligible effect on the performance.

Table 6 (BZR_MD) shows significant changes in the performance for all our vectorizations (EMP-Betti, EMP-Silhouette and EMP-Entropy). In Figure 4a, we also see in half of the models, one order is better than the other, while in the other half of the models, the opposite is true. On the other hand, we observe that for the REDDIT-BINARY dataset, the order is not as important for these functions (Table 7 & Figure 4b) as the change in performance is not as significant as in BZR_MD.

Table 6: Graph Dataset BZR_MD. Impact on classification accuracy when modifying the filtration ordering. $S_{\mathcal{V}}$ and $W_{\mathcal{E}}$ denotes sublevel filtration on nodes and weight filtration on edges, respectively. The statistical significance, when one order is better that the other, is shown at three levels (0.1, 0.05, 0.01) and the best overall result of each order filtration is in **bold** font.

| | Order | none | $f_{\mathcal{V}}$ | $f_{\mathcal{E}}$ | $f_{\mathcal{V}} + f_{\mathcal{E}}$ |
|---|---|---|---|---|---|
| none | - | - | $67.124 \pm 1.959$ | $58.714 \pm 0.520$ | $67.642 \pm 1.724$ |
| none | - | - | $67.124 \pm 1.959$ | $58.714 \pm 0.520$ | $67.642 \pm 1.724$ |
| EMP-B | $S_{\mathcal{V}} \rightsquigarrow W_{\mathcal{E}}$ | $71.414 \pm 1.638$ | $71.446 \pm 1.166$ | $71.351 \pm 1.107$ | $71.610 \pm 1.125$ |
| $H_0$ | $W_{\mathcal{E}} \rightsquigarrow S_{\mathcal{V}}$ | $76.192 \pm 1.188^{***}$ | $\textbf{\textit{76.363}} \pm \textbf{\textit{0.861}}^{***}$ | $75.804 \pm 1.025^{***}$ | $75.809 \pm 0.989^{***}$ |
| EMP-B | $S_{\mathcal{V}} \rightsquigarrow W_{\mathcal{E}}$ | $69.047 \pm 1.092$ | $69.337 \pm 0.833$ | $69.114 \pm 0.711$ | $69.798 \pm 0.845$ |
| $H_1$ | $W_{\mathcal{E}} \rightsquigarrow S_{\mathcal{V}}$ | $69.094 \pm 1.231$ | $70.196 \pm 0.860^{**}$ | $69.648 \pm 0.894$ | $70.296 \pm 0.973$ |
| EMP-B | $S_{\mathcal{V}} \rightsquigarrow W_{\mathcal{E}}$ | $72.789 \pm 1.544$ | $72.946 \pm 1.502$ | $72.848 \pm 1.286$ | $73.204 \pm 1.568$ |
| $H_0 + H_1$ | $W_{\mathcal{E}} \rightsquigarrow S_{\mathcal{V}}$ | $75.156 \pm 0.647^{***}$ | $74.996 \pm 0.667^{***}$ | $75.160 \pm 0.821^{***}$ | $75.055 \pm 0.926^{***}$ |
| EMP-E | $S_{\mathcal{V}} \rightsquigarrow W_{\mathcal{E}}$ | $71.967 \pm 1.091$ | $72.129 \pm 1.335$ | $71.803 \pm 1.253$ | $72.198 \pm 1.280$ |
| $H_0$ | $W_{\mathcal{E}} \rightsquigarrow S_{\mathcal{V}}$ | $73.274 \pm 0.963^{**}$ | $74.285 \pm 1.206^{***}$ | $73.314 \pm 1.258^{**}$ | $74.812 \pm 1.419^{***}$ |
| EMP-E | $S_{\mathcal{V}} \rightsquigarrow W_{\mathcal{E}}$ | $72.485 \pm 1.209^{***}$ | $72.452 \pm 1.121^{***}$ | $72.326 \pm 0.725^{***}$ | $72.230 \pm 1.330^{***}$ |
| $H_1$ | $W_{\mathcal{E}} \rightsquigarrow S_{\mathcal{V}}$ | $67.097 \pm 1.364$ | $65.853 \pm 1.457$ | $66.274 \pm 1.370$ | $66.246 \pm 1.273$ |
| EMP-E | $S_{\mathcal{V}} \rightsquigarrow W_{\mathcal{E}}$ | $75.498 \pm 1.338$ | $75.819 \pm 1.209^{***}$ | $75.133 \pm 1.357^{**}$ | $75.303 \pm 1.303^{***}$ |
| $H_0 + H_1$ | $W_{\mathcal{E}} \rightsquigarrow S_{\mathcal{V}}$ | $74.985 \pm 1.112$ | $73.410 \pm 1.378$ | $73.841 \pm 1.200$ | $73.053 \pm 1.227$ |
| EMP-S | $S_{\mathcal{V}} \rightsquigarrow W_{\mathcal{E}}$ | $69.191 \pm 1.138$ | $69.717 \pm 0.859$ | $69.224 \pm 0.960$ | $69.417 \pm 1.036$ |
| $H_0$ | $W_{\mathcal{E}} \rightsquigarrow S_{\mathcal{V}}$ | $75.926 \pm 1.051^{***}$ | $76.320 \pm 0.952^{***}$ | $75.927 \pm 0.597^{***}$ | $76.187 \pm 0.713^{***}$ |
| EMP-S | $S_{\mathcal{V}} \rightsquigarrow W_{\mathcal{E}}$ | $74.067 \pm 0.534^{***}$ | $74.428 \pm 0.723^{***}$ | $74.432 \pm 1.003^{***}$ | $74.659 \pm 0.919^{***}$ |
| $H_1$ | $W_{\mathcal{E}} \rightsquigarrow S_{\mathcal{V}}$ | $70.232 \pm 1.211$ | $69.903 \pm 1.481$ | $70.065 \pm 1.347$ | $70.068 \pm 1.141$ |
| EMP-S | $S_{\mathcal{V}} \rightsquigarrow W_{\mathcal{E}}$ | $77.658 \pm 0.692^{***}$ | $\textbf{\textit{78.025}} \pm \textbf{\textit{0.656}}^{***}$ | $77.794 \pm 0.496^{***}$ | $77.861 \pm 0.802^{***}$ |
| $H_0 + H_1$ | $W_{\mathcal{E}} \rightsquigarrow S_{\mathcal{V}}$ | $74.760 \pm 1.394$ | $75.286 \pm 1.054$ | $75.028 \pm 1.116$ | $75.128 \pm 1.099$ |

Table 7: Graph Dataset REDDIT-BINARY. Impact on classification accuracy when modifying the filtration ordering. $S_\mathcal{V}$ and $W_\mathcal{E}$ denotes sublevel filtration on nodes and weight filtration on edges, respectively. The statistical significance, when one order is better that the other, is shown at three levels (0.1, 0.05, 0.01) and the best overall result of each order filtration is in **bold** font.

| | Order | none | $f_\mathcal{V}$ | $f_\mathcal{E}$ | $f_\mathcal{V} + f_\mathcal{E}$ |
|---|---|---|---|---|---|
| none | - | - | $89.680 \pm 0.300$ | $79.590 \pm 0.394$ | $90.125 \pm 0.277$ |
| none | - | - | $89.680 \pm 0.300$ | $79.590 \pm 0.394$ | $90.125 \pm 0.277$ |
| EMP-B | $S_\mathcal{V} \rightsquigarrow W_\mathcal{E}$ | $88.925 \pm 0.168$ | $90.400 \pm 0.145$ | $89.145 \pm 0.193$ | $90.520 \pm 0.204$ |
| $H_0$ | $W_\mathcal{E} \rightsquigarrow S_\mathcal{V}$ | *$90.005 \pm 0.196$\*\*\** | *$90.825 \pm 0.136$\*\*\** | *$90.135 \pm 0.161$\*\*\** | ***$90.925 \pm 0.133$\*\*\**** |
| EMP-B | $S_\mathcal{V} \rightsquigarrow W_\mathcal{E}$ | $80.715 \pm 0.235$ | $87.090 \pm 0.217$ | $84.620 \pm 0.160$ | $88.225 \pm 0.280$ |
| $H_1$ | $W_\mathcal{E} \rightsquigarrow S_\mathcal{V}$ | $80.880 \pm 0.335$ | *$88.300 \pm 0.334$\*\*\** | *$85.030 \pm 0.363$\*\*\** | *$88.895 \pm 0.323$\*\*\** |
| EMP-B | $S_\mathcal{V} \rightsquigarrow W_\mathcal{E}$ | $89.970 \pm 0.309$ | *$90.945 \pm 0.184$\*\** | $90.025 \pm 0.234$ | ***$91.025 \pm 0.221$\*\**** |
| $H_0 + H_1$ | $W_\mathcal{E} \rightsquigarrow S_\mathcal{V}$ | $90.020 \pm 0.200$ | $90.650 \pm 0.270$ | $90.125 \pm 0.154$ | $90.770 \pm 0.198$ |
| EMP-E | $S_\mathcal{V} \rightsquigarrow W_\mathcal{E}$ | $88.140 \pm 0.265$ | $89.875 \pm 0.185$ | $88.400 \pm 0.210$ | $89.945 \pm 0.177$ |
| $H_0$ | $W_\mathcal{E} \rightsquigarrow S_\mathcal{V}$ | *$89.695 \pm 0.157$\*\*\** | *$90.245 \pm 0.147$\*\*\** | *$89.850 \pm 0.199$\*\*\** | *$90.450 \pm 0.186$\*\*\** |
| EMP-E | $S_\mathcal{V} \rightsquigarrow W_\mathcal{E}$ | *$79.850 \pm 0.365$\*\*\** | $87.670 \pm 0.303$ | *$84.600 \pm 0.214$\*\*\** | *$88.640 \pm 0.156$\*\*\** |
| $H_1$ | $W_\mathcal{E} \rightsquigarrow S_\mathcal{V}$ | $74.560 \pm 0.308$ | $87.545 \pm 0.196$ | $83.755 \pm 0.239$ | $88.350 \pm 0.196$ |
| EMP-E | $S_\mathcal{V} \rightsquigarrow W_\mathcal{E}$ | $89.285 \pm 0.234$ | $90.010 \pm 0.234$ | $89.240 \pm 0.202$ | $90.045 \pm 0.221$ |
| $H_0 + H_1$ | $W_\mathcal{E} \rightsquigarrow S_\mathcal{V}$ | *$90.015 \pm 0.192$\*\*\** | *$90.235 \pm 0.243$\** | *$90.130 \pm 0.200$\*\*\** | *$90.275 \pm 0.171$\*\** |
| EMP-S | $S_\mathcal{V} \rightsquigarrow W_\mathcal{E}$ | $86.150 \pm 0.261$ | $87.925 \pm 0.162$ | $86.940 \pm 0.206$ | $88.415 \pm 0.241$ |
| $H_0$ | $W_\mathcal{E} \rightsquigarrow S_\mathcal{V}$ | *$89.380 \pm 0.117$\*\*\** | *$90.345 \pm 0.205$\*\*\** | *$89.545 \pm 0.207$\*\*\** | *$90.455 \pm 0.156$\*\*\** |
| EMP-S | $S_\mathcal{V} \rightsquigarrow W_\mathcal{E}$ | *$77.580 \pm 0.186$\*\*\** | $85.435 \pm 0.332$ | $81.580 \pm 0.222$ | $86.960 \pm 0.346$ |
| $H_1$ | $W_\mathcal{E} \rightsquigarrow S_\mathcal{V}$ | $75.165 \pm 0.367$ | *$86.775 \pm 0.292$\*\*\** | *$83.125 \pm 0.415$\*\*\** | *$87.295 \pm 0.325$\*\** |
| EMP-S | $S_\mathcal{V} \rightsquigarrow W_\mathcal{E}$ | $86.790 \pm 0.385$ | $88.250 \pm 0.266$ | $87.410 \pm 0.315$ | $88.590 \pm 0.377$ |
| $H_0 + H_1$ | $W_\mathcal{E} \rightsquigarrow S_\mathcal{V}$ | *$89.715 \pm 0.279$\*\*\** | *$90.495 \pm 0.248$\*\*\** | *$89.780 \pm 0.337$\*\*\** | *$90.580 \pm 0.217$\*\*\** |

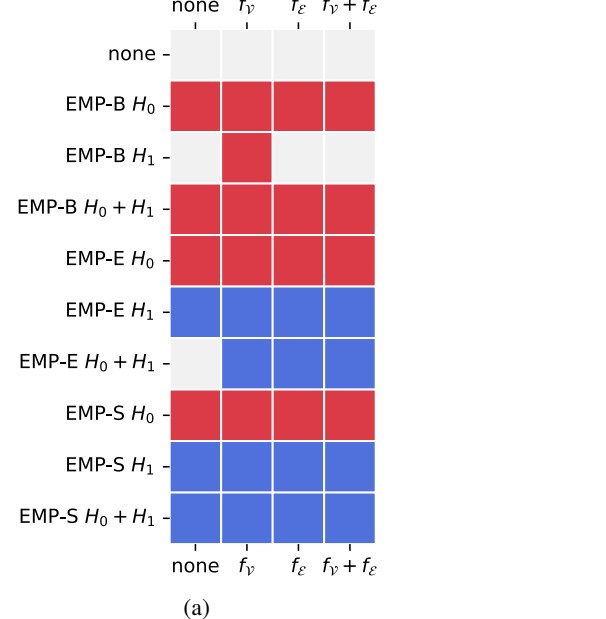 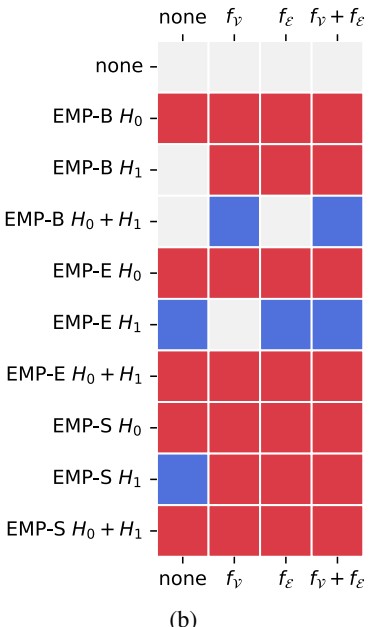

Figure 4: Visual summary of the impact on classification accuracy when modifying the filtration ordering, for each combination of features and EMP summaries. Whenever statistical hypothesis support a winner, the corresponding cell is coloured either as **blue** ($S_\mathcal{V} \rightsquigarrow W_\mathcal{E}$), or **red** ($W_\mathcal{E} \rightsquigarrow S_\mathcal{V}$). (a) BZR_MD. (b) REDDIT-BINARY.

