# OpenReview forum: "EMP: Effective Multidimensional Persistence for Graph Representation Learning"
_ICLR.cc/2023/Conference — Submitted to ICLR 2023_

### Official Review · Reviewer_xbqM · 2022-10-18

**Confidence:** 5
**Correctness:** 4
**Technical Novelty And Significance:** 2
**Empirical Novelty And Significance:** 3
**Recommendation:** 6

**Clarity, Quality, Novelty And Reproducibility:**

## Clarity

The paper is written very clearly for the most part. As a contribution that straddles different worlds, it would be useful to provide a more intuitive on some of the topological concepts because a machine learning audience may not be familiar with them. Here are additional suggestions:

- Section 3, in particular Section 3.1, may benefit from a more intuitive description or potentially a running example (which would also be useful when going to the multidimensional case).

- When comparing to other methods in Section 5.2 (a section whose experimental depth I very much appreciate!), please provide some details on the parameters and the setup of the methods. For instance, which internal kernel was used for the CSM kernel? Which type of filtration was used for the filtration curves? etc. (it would be sufficient to provide a brief paragraph in the appendix or, space permitting, in the main paper)

- Concerning the terminology of the paper, I would prefer "filtration function" instead of "filtering function"

- Some of the description in Section 4.1 seem slightly redundant: the second paragraph of the section rehashes the same concepts as the first one. Consider rewriting it. Similarly, the examples in Section 4.2, while providing a comprehensive overview, could also be (partially) relegated to the appendix. Since multiparameter persistence images and persistence landscapes already exist, the discussion does not provide additional insights for readers unfamiliar with concepts from topological data analysis. I would suggest using this space to provide a worked example of the proposed slicing approach, with showcasing only a single visualisation or "featurisation" method, if possible.

- Section 4.3 should come *before* introducing examples of featurisation techniques. This is a very central result to the current paper, and it should be discussed in more detail.

- To what extent does the proposed method work for simplicial complexes as well? I see no direct obstacles; all that appears to be required is a way to enumerate the filtration steps (which should always work in practice). Is there a particular reason (other than the proposed application) for focusing on graphs?

- The first paragraph of Section 5.2 is slightly redundant; most of the information is already discussed in Section 5.1

## Quality

I appreciate the use of ablation studies throughout the paper. In terms of the theoretical contributions of the paper, please find below some suggestions to further improve the quality of the paper.

- One of the main things I have yet to "grok" is the implications of restricting the filtration to the type of slices shown in the paper. I would suggest to provide a worked example here. When introducing bi-filtrations for the first (?) time, [Carlsson and Zomorodian](https://link.springer.com/article/10.1007/s00454-009-9176-0) used an example comprising curvature and a distance parameter, for instance. As far as I understand, this approach only uses a "horizontal" slice. Would it be possible to quantify how much information is lost in the worst case by this approach?

- Consider discussing the implications of changing the order of filter functions in more detail. It would be best to show a brief example that showcases the implication of a specific ordering.

## Novelty

My main issue as outlined above concerns the delineation from existing work. An improved discussion of the issues with multidimensional persistence would definitely strengthen the paper. Since multiparameter persistence has been around since (at least) 2009, when Carlsson and Zomorodian [wrote their seminal paper](https://link.springer.com/article/10.1007/s00454-009-9176-0), I would suggest to tone down statements such as "[The] MP approach does not have theoretical foundations yet [...]". They are just not accurate. I understand that multiparameter persistence is even more complex than the single-parameter approach, but we as a community have by now a sufficiently good understanding of the background.

## Reproducibility

The paper should be reproducible by a skilled graduate student with some prior exposure to topological methods. As far as I can tell, all important algorithmic choices are provided to reproduce the presented methods. While some of the experimental details (in particular the setup of comparison partners) could be extended, this is not an obstacle towards running the overall pipeline.

## Minor issues

- There are some minor issues with missing articles in the text. For instance, in Section 3.1, it should be "*a* clique complex", "*the* $k$th persistence" diagram", and "for *the* sake of notation, we skip". Another pass over the text might be beneficial.

- In the abstract: "convert them" --> "converts them"

- Please employ `\citep` when referring to parenthetical citations, and `\citet` when referring to in-text citations. Right now, *all* citations are used in the textual form, making me stumble over them while reading.

- When discussing topology-based approaches for graph classification, consider citing [Hofer et al. (2020)](https://proceedings.mlr.press/v119/hofer20b.html), [Bodnar et al. (2022)](https://arxiv.org/abs/2202.04579), and  [Horn et al. (2022)](https://openreview.net/forum?id=oxxUMeFwEHd) as recent approaches combining topological features and deep learning.

- "As such" is being used twice in close succession  in the discussion

- Please re-check the bibliography for consistency issues. There are several problems with capitalisation ("riemannian" instead of "Riemannian", "betti" instead of "Betti", etc.).

- Minor nitpicky comment: consider rephrasing some more uncommon formulations such as "This step is pretty standard".

- "$\phi$ holds such a stability inequality" --> "$\phi$ satisfies such a stability property/inequality"



**Strength And Weaknesses:**

The primary strength of the paper lies in providing a **novel algorithm for slicing multiparameter persistence modules**. While demonstrated for bi-filtrations on graphs, the method should also easily extend to simplicial complexes, thus opening the door towards making computational topology approaches applicable in such a scenario. The secondary aspect of the paper I appreciate is the use of featurisation techniques from persistent homology to solve graph classification tasks. This showcases the utility of persistent homology techniques in machine learning scenarios.

The primary weakness of the paper is the **missing delineation to prior work**, making it hard to judge the contribution. Specifically, I would like to see a more in-depth discussion of the [*sliced barcode* technique pioneered by Lesnick and Wright](https://arxiv.org/abs/1512.00180). The main contribution of the paper at hand appears to be a specific type of slicing, namely a "horizontal" slicing through a bi-persistence module, raising questions of (a) why this specific slicing technique was selected, and (b) whether the proposed method does not amount to an algorithm that essentially featurises a multiparameter persistence module along a given path (I understand that there are minor differences in the way the filtration is calculated; essentially, this method proposes to calculate *multiple* slices, but the question about the general principle still remains). To this end, the cited paper by Carrière and Blumberg should be discussed in more detail. To the best of my knowledge, they were the first to discuss such slicing techniques in a machine learning context. While I am happy to be proven wrong, I think it would be beneficial for the paper if the main contribution were to be (re)phrased in such a manner.

A secondary aspect of the paper that requires some clarification concerns the **experimental results**: here, the `EMP` method uses a variety of different summary functions and featurisation strategies. Since these techniques are of different expressive powers, it would be crucial to understand *which* precise method was chosen for a given data set (or even for a specific split). If I understand the experiments correctly, different splits of the same data set could conceivably make use of different featurisation strategies, giving `EMP` a certain advantage. If this is correct, it should at least be briefly discussed; the discussion of the precise featurisation strategy for a given data set may also be of independent interest (I am imaging that "simpler" methods such as multiparameter Betti curves probably fare less well).


**Summary Of The Paper:**

This paper presents a novel topology-driven approach for feature generation in data sets. The paper is fundamentally built upon the paradigm of persistent homology, a method for obtaining multi-scale topological features of complex data sets such as point clouds and graphs. Since "ordinary" persistent homology is restricted to providing topological features according to a *single* scale parameter, the paper leverages multidimensional persistence theory to enable the use of multiple such parameters. This permits viewing a data set under different data lenses, such as a distance parameter versus a density parameter. Multidimensional persistence theory being mathematically more cumbersome that the single-parameter approach, the paper introduces a smart "slicing" strategy, turning any multi-parameter into a set of single-parameter approaches.

In addition, the paper shows how existing topological featurisation strategies, i.e. algorithms for calculating topological features from data, can be recast into such a multi-parameter setting. Experiments in graph classification serve as a culmination of the research directions outlined in the paper.


**Summary Of The Review:**

This is an exciting contribution to the emerging field of computational topology. While the paper in its present form suffers from some issues, in particular with respect to the delineation to existing work and the clarity of the experimental setup, I do believe that these issues can be overcome within the revision cycle. Thus, I tend to conditionally endorse this paper for an eventual presentation at the conference, provided that the issues outlined above are sufficiently well addressed in a revision.

---

### Official Review · Reviewer_mkf8 · 2022-10-18

**Confidence:** 3
**Correctness:** 4
**Technical Novelty And Significance:** 3
**Empirical Novelty And Significance:** 3
**Recommendation:** 5

**Clarity, Quality, Novelty And Reproducibility:**

## Clarity

The paper is not easy to read. It is hard to catch the concept behind the two main tools. There is an illustration in the appendix that would be better in the main article. Not sure that the most machine learning community could read this paper.

## Quality

The work is good, but difficult to apprehend.

## Novelty

The proposed framework relies on existing tools. The main contribution is more about new combination of these tools that has not been done before. The stability guarantee is also novel.

## Reproducibility

The code is available. However, I am not sure that the framework is directly reproducible from the paper.

**Strength And Weaknesses:**

## Strength

The propose topological data analysis is interesting on several points
1. It is fairly general as it allow data attributes on nodes.
2. It is able to catch complex structure that may appear with graphs.
3. It is not much sensitive to structural noise.
4. The filtration tools can be dedicated to some tasks helping to catch hidden features.
5. There is a stability guarantee.

## Weaknesses

Sadly this paper has several strong weaknesses.
1. It is very hard to follow for someone not familiar with persistent homology.
2. It is not fit for high dimensional features on the nodes.
3. The order of the filters is important, the authors say that one can use the importance of the features. This is something that could be hard to find.
4. The experiments only include classical GNN, there exists now more powerful GNN.

**Summary Of The Paper:**

This paper proposes a mathematical framework for working with graphs. The framework is based on two main tools: the multiparameter persistence and persistent homology. This build a framework for topological data analysis that is able to use the data attributes on the nodes. The experiments show promising results with strong improvement on some data sets.

**Summary Of The Review:**

This paper presents a topological data analysis framework for graphs based on persistent homology and multipersistence. This lead to a stable framework for graph learning and the experiments shows promising results. However the tools are not common in the ML community thus, the paper is hard to read and understand. Furthermore some weakness about the order of filters and the dimension of features reduce the interest of the framework.

---

### Official Review · Reviewer_t3P2 · 2022-10-24

**Confidence:** 4
**Correctness:** 3
**Technical Novelty And Significance:** 2
**Empirical Novelty And Significance:** 3
**Recommendation:** 5

**Clarity, Quality, Novelty And Reproducibility:**

This is a well-written description of the proposed methodology. On the other hand, it is vague and overstated in its subject matter, and the novelty is accordingly unclear.

**Strength And Weaknesses:**

Strength

- TDA is very powerful, but does not work well enough when there are multiple meaningful filtrations. Graph data is a target for many such examples, and this method proposes a method for handling multiple filtrations. This is very meaningful.
- Experiments show the effect on the benchmarks of the graph data.

Weakness

- The difficulty in handling multiple filtrations is that there are multiple paths between two points on the filtration parameter space, and not all paths may lead to the same state. Another problem is that, depending on the definition of multiple filtration, it may be difficult to define birth and death points, and the persistent diagram may be complicated by curves instead of points in space. The proposed method cannot be said to overcome the above. It avoids the above issues for a limited number of targets with limited conditions.
- It is admitted that the proposed method is one of the realistic methods to handle multiple filtration. However, it cannot handle any filtration. For example, the proposed method is not applicable to rips filtration and weighed rips filtration for point clouds, because one filtration would make it not a point cloud anymore. Although the authors write that the method can be applied to examples other than those in this paper, a clear discussion should be made regarding the conditions. At least the previously applied filtration should not change the data format.
- There have been previous efforts regarding the concept of this paper. There is at least the following (though this is not a paper).
https://patentimages.storage.googleapis.com/80/21/7f/3d5033afb8ef36/US11024022.pdf
Taking the above into account, the novelty of this paper may be its application to limited graphical data. Clearly discuss where the novelty lies.
- It is possible to find value in the method as a method for the data that is the subject of the experiment, but the comparison is never state-of-the-art. For example, U2GNN (https://arxiv.org/pdf/1909.11855.pdf), DUGNN (https://arxiv.org/abs/1909.10086), FactorGCN (https://arxiv.org/abs/2010.05421 ), etc. should also be compared.

**Summary Of The Paper:**

This paper proposes a method for dealing with multi-dimensional persistetnce in TDA, i.e., the presence of multiple filtration parameters. In particular, it proposes a method for graph data and provides a vectorization method for machine learning from the information obtained from the proposed method. Finally, the effectiveness of the proposed method for graph data is evaluated for multiple data sets.

**Summary Of The Review:**

The awareness of the issue is important and the effect on the subject matter being experimented with is acknowledged. However, it is presumed that the extent to which it is adaptable and effective is limited. This point seems to have been overstated. In addition, the concept of the methodology itself has been around for some time. Even when considered as a method to improve the performance of what is being experimented with, it lacks comparison with the state-of-the-art.

**Conclusion following discussion with the authors**

The authors' claim was clarified that the most core ideas about multidimensional TDA are existing, but that they give specific computational methods and vectorizations for graphs. It is also shown to be a framework that can be used in various filtration for graphs, and concrete effects are also shown. Although users will need to find the appropriate settings for their specific data, this is a good primer for those who want to apply TDA to their graphs. I hope that the authors will revise their statement, as it still contains some exaggerated misrepresentations of their claims. I will revise the score to reflect the authors' clarification of their claims, but the score will take into account the impact of their proposal.

---

### Decision · Program_Chairs · 2023-01-20

**Decision:**

Reject

**Justification For Why Not Higher Score:**

The paper has some weaknesses as pointed out by reviewers. First, the relationship between the proposed method and the previous work is not clear. Moreover, the experimental section can be further improved.

**Justification For Why Not Lower Score:**

N/A

**Metareview: Summary, Strengths And Weaknesses:**

In this paper, the authors propose a method for dealing with multi-dimensional persistence in topological data analysis (TDA). The strength of the paper is to provide a novel algorithm for slicing multiparameter persistence modules. Reviewers agree that the proposed method is interesting. However, the paper has some weaknesses as pointed out by reviewers. First, the relationship between the proposed method and the previous work is not clear. Moreover, the experimental section can be further improved. We discussed the paper carefully during the discussion period based on the author's rebuttal. However, we concluded that it needs a major revision before publication.

Thus, I encourage the authors to revise the paper based on the reviewer's comments and resubmit it to a future venue.

**Summary Of Ac-Reviewer Meeting:**

N/A